# Determination of the vertical distribution of in-cloud particle shape using SLDR mode 35-GHz scanning cloud radar

Audrey Teisseire[1], Patric Seifert[1], Alexander Myagkov[2], Johannes Bühl[1], and Martin Radenz[1]

[1]Leibniz Institute for Tropospheric Research, Permoserstraße 15, 04318 Leipzig, Germany
[2]RPG Radiometer Physics GmbH, Werner-von-Siemens-Str. 4, 53340 Meckenheim, Germany

**Correspondence:** Audrey Teisseire (teisseire@tropos.de)

**Abstract.** In this study we present an approach that uses the polarimetric variable SLDR (Slanted Linear Depolarization Ratio) from a scanning polarimetric cloud radar MIRA-35 in the SLDR configuration, to derive the vertical distribution of particle shape (VDPS) between top and base of mixed-phase cloud systems. The polarimetric parameter SLDR was selected for this study due to its strong sensitivity to shape and low sensitivity to the wobbling effect of particles at different antenna elevation angles. For the VDPS method, elevation scans from $90°$ to $30°$ elevation angle were deployed to estimate the vertical profile of the particle shape by means of the polarizability ratio, which is a measure of the density-weighted axis ratio. Results were obtained by retrieving the best fit between observed SLDR from $90°$ to $30°$ elevation angle and respective values simulated with a spheroidal scattering model. The applicability of the new method is demonstrated by means of three case studies of isometric, columnar and plate-like hydrometeor shapes, respectively, which were obtained from measurements at the Mediterranean site of Limassol, Cyprus. The identified hydrometeor shapes are demonstrated to fit well to the cloud and thermodynamic conditions which prevailed at the times of observations. A fourth case study demonstrates a scenario where ice particle shapes tend to evolve from a pristine state at cloud top toward a more isometric shape or less dense particles at cloud base. Either aggregation or riming processes contribute to this vertical change of microphysical properties. The new height-resolved identification of hydrometeor shape and the potential of the VDPS method to derive its vertical distribution are helpful tools to understand complex processes such as riming or aggregation, which occur particularly in mixed-phase clouds.

## 1 Introduction

In the troposphere, a rich variety of cloud types exists, which are formed by characteristic microphysical processes. The structure of clouds is in general determined by the complex interaction of water vapor, ice, liquid droplets, vertical air motion and aerosol particles, acting as cloud condensation nuclei (CCN) or ice nucleating particles (INP) (Pruppacher and Klett, 1997; Morrison et al., 2012; Ansmann et al., 2019). While in warm clouds the collision-coalescence process is the primary process responsible for the formation of precipitation, the situation is more complicated in ice-containing clouds having temperatures between $-40°C$ and $0°C$. In this temperature range, the coexistence of supercooled liquid water and ice is possible. Thus, in these mixed-phase clouds, multiple cloud microphysical processes are intertwined as they contain a three-phase colloidal system consisting of water vapor, ice particles and supercooled liquid droplets (Korolev et al., 2017). The initial partitioning

between the ice and liquid water is determined by the CCN and INP reservoir and represents the prevalent conditions for secondary ice formation processes, riming and aggregation (Solomon et al., 2018; Fan et al., 2017), which are greatly involved in the precipitation transition in mixed-phase clouds.

Observation of the hydrometeor habit is a possible way to study cloud formation and precipitation because particle shape can be considered a fingerprint of crucial processes, including crystal growth, evaporation rate, ice crystal fall speed, and cloud radiative properties (e.g., Avramov and Harrington, 2010). Shape allows to distinguish pristine ice crystals from hydrometeors which have grown via aggregation or riming processes and can be considered as a tracer of the different processes contributing to the evolution of a cloud system. The overall structure of ice crystals grown in air can be classified into plate-like and columnar shapes as a function of temperature between $-40°$ and $0°C$. Bühl et al. (2016) and Myagkov et al. (2016a) showed that primary ice formation dominates in thin layers of stratiform or mixed-phase clouds of a geometrical thickness $\leq 350\,\mathrm{m}$ , as growth processes in these thin clouds are constrained (Fukuta and Takahashi, 1999). In such cloud systems and conditions of liquid water saturation, the shape of ice crystals is thus related directly to the environmental temperature (Myagkov et al., 2016b). However, further complexity can be expected when the cloud systems become deeper and when the thermodynamic structure is less well defined as in single-layer stratiform mixed-phase clouds. Techniques which allow detecting the hydrometeor shape have high potential to contribute additional capabilities for the monitoring of cloud systems, to expand the understanding of the involved microphysical properties, and to support the improvement of the representation of these processes in numerical models. A way to discriminate different hydrometeor populations is the separation of peaks in cloud radar Doppler spectra (Radenz et al., 2019; Kalesse et al., 2019; Luke et al., 2021) using observations of ground-based cloud radar. However, this technique is limited, e.g., with respect to atmospheric turbulence, which broadens the spectra and makes the detection and separation of peaks difficult or even impossible. Moreover, hydrometeors with similar terminal fall velocities (for example drizzle and small ice) cannot be distinguished in the Doppler spectrum. In this case, it is possible to have a look at the Doppler spectra of polarimetric parameters such as LDR or SLDR to confirm in which spectral mode the crystals are present.

Polarimetric cloud radar techniques have been shown to be valuable tools for the qualitative detection of ice crystal shape (Matrosov et al., 2001; Reinking et al., 2002; Matrosov et al., 2005). Matrosov et al. (2012) demonstrated an approach where they associated measurements of Slanted Linear Depolarization Ratio (SLDR) mode scanning cloud radar to visual observations of ice crystal habits during a precipitation event. While their study demonstrates well the relationship between SLDR signatures and particle shape, it did not yet allow to quantify the particle shape directly based on the measurements. Such an approach has been presented by Myagkov et al. (2016a), who succeeded in predicting the particle shape and orientation based on hybrid-mode scanning cloud radar observations by means of the two quantitative parameters polarizability ratio and degree of orientation, respectively. Myagkov et al. (2016a) have shown that existing backscattering models, assuming the spheroidal approximation of cloud scatters, can be applied to establish a link between a set of measured polarimetric variables and the polarizability ratio. Polarizability ratio is a parameter defined by the geometric aspect ratio of particles and their refractive index. For ice particles the refractive index is almost a linear function of their apparent ice density. Note, that it is not directly possible to infer the aspect ratio and the apparent ice density from the polarizability ratio. However, since the polarizability ratio depends on both variables, it can be used to track the evolution of the ice particles from pristine state to aggregates and

rimed particles in observational studies. Polarizability ratio profiles are also valuable for modeling studies since the profiles can be used to constrain microphysical processes of ice growth. The first attempt to utilize polarizability ratios to improve ice characterization in models was recently done by Welss et al. (2023). Based on polarizability ratios the authors have updated the ice growth characterization for the explicit habit prediction in the the Lagrangian super-particle ice microphysics model Mc-Snow developed by German Weather Service (DWD, Brdar and Seifert (2018)). Even though developed for SLDR mode and Simultaneous Transmit Simultaneous Receive (STSR, hybrid)-mode cloud radars, applicability of the shape and orientation estimation retrieval was originally demonstrated only for a STSR-mode scanning 35-GHz cloud radar, based on observations of stratiform cloud layers during the one-month field campaign Analysis of Composition of Clouds with Extended Polarization Techniques (ACCEPT, Myagkov et al. (2016a)).

Even though the number of scanning STSR-mode cloud radars has been continuously growing in Europe, a number of measurement sites within ACTRIS (the Aerosol, Clouds and Trace Gases Research Infrastructure) are equipped with scanning LDR radars (Madonna et al., 2013; Löhnert et al., 2015; Tetoni et al., 2022). Such radars can be modified to the SLDR mode with relatively low efforts and investments, and as a result can provide long-term observational datasets for retrieving the polarizability ratio of ice-containing clouds in different climatic zones. Therefore, the main goal of this study is to derive the vertical distribution of particle shape in clouds using the spheroidal scattering model developed by Myagkov et al. (2016a) for application to regular long-term observations of a SLDR-mode 35-GHz scanning cloud radar. We introduce a simplified and versatile version of the original STSR-mode approach by concentrating on the retrieval of the polarizability ratio, as we consider this parameter to be more relevant for the investigation of cloud microphysical processes in comparison to the degree of orientation. This paper aims on demonstrating the ability of the Vertical Distribution of Particle Shape (VDPS) method, to characterize particle properties using data with a newly configured SLDR-mode 35-GHz cloud radar which was deployed in the Cyprus Clouds, Aerosols and pRecipitation Experiment (CyCARE, Ansmann et al. (2019)) field campaign in Limassol, Cyprus. We also illustrate that a profile of the derived polarizability ratio can be potentially used to detect microphysical processes affecting the evolution of ice particles in deep precipitating clouds. In Section 2, instrumentation, campaign setup and the polarimetric parameter SLDR will be described. The VDPS method will be introduced in Section 3 and an evaluation of the VDPS method is presented in Section 4. Case studies showing isometric particles, columnar crystals and, plate-like crystals will be discussed, and a fourth case study showing a transformation in shape of particles from cloud top to cloud base will be presented to demonstrate the potential of the VDPS method to detect and describe microphysical transformation processes. In Section 5, we will elaborate on the advantages and limits of this new algorithm as well as on possible future improvements.

## 2  Dataset

### 2.1  SLDR mode 35-GHz cloud radar MIRA-35

The central instrument for the present study is a modified version of the 35-GHz cloud radar Mira-35, which is operated in SLDR mode. MIRA-35 in general is a dual-polarization (LDR-mode) radar which emits linearly polarized radiation through the co-channel, while the returned signals are received in both the co- and cross-channels. The SLDR mode cloud radar

**Table 1.** Technical characteristics of MIRA-35 SLDR-mode cloud radar during the deployment in the CyCARE campaign in Limassol, Cyprus.

| Parameters | Values |
|---|---|
| Pulse power | 30 kW |
| Pulse length | 208 ns |
| Pulse repetition frequency | 7500 Hz |
| Elevation angle velocity | 0.5 deg s$^{-1}$ |
| nFFT points | 512 |
| Number of range gates | 498 |
| Number of spectral averages | 15 |
| Integration time | 1 s |
| Range resolution | 31.18 m |
| Reflectivity sensitivity (1 s averaging at range=5 km) | −48 dBZ |
| co-cross-channel isolation | −35 dB |

was implemented based on the conventional Linear Depolarization Ratio (LDR) mode by $45°$ rotation of the antenna system around the emission direction. While numerous polarimetric configurations of radar systems exist (Bringi and Chandrasekar, 2001, Ch. 6), the LDR mode is currently the most common one amongst cloud radars. The properties of the standard LDR mode Mira-35 are elaborated in detail in Görsdorf et al. (2015). The technical characteristics of MIRA-35 used in the CyCARE campaign in Limassol, Cyprus, are given in the Table 1. Standard vertical-stare LDR-mode allows only to discriminate between hydrometeors with an isometric intersection and with a columnar intersection (Bühl et al., 2016). I.e., aggregates cannot be separated from generally horizontally oriented plate-like particles in vertical-stare mode because their scattering intersections appear to be similar. In order to optimize the Mira-35 cloud radar for improved measurements of hydrometeor shape and orientation, two modifications were applied to the standard setup as it is described by Görsdorf et al. (2015). First, the cloud radar was mounted onto a positioner platform which allows for a freely definable position of the radar within a half sphere given by $360°$ of azimuth and $180°$ of elevation. The second modification addresses a $45°$ rotation of the antenna around the emission direction. This operation mode, in general defined as SLDR mode, has specific advantages in studies of the intrinsic relationship between the polarimetric signature of the particle shape and radar elevation angle. In contrast to the standard LDR mode, variations in the orientation of hydrometeors only have small effects on the measured SLDR, even at low elevation angles (Matrosov, 1991). In turn, SLDR in vertical pointing mode (elevation = $90°$) is similar to the LDR observed with standard Mira-35 systems. This behavior is also of advantage because it ensures direct comparability to other standard LDR-mode radars in vertical-pointing measurements. In the framework of the presented study, the radar was steered toward geographic south direction ($180°$ azimuth angle) and performed range-height-indicator (RHI) scans from $90°$ (zenith-pointing)

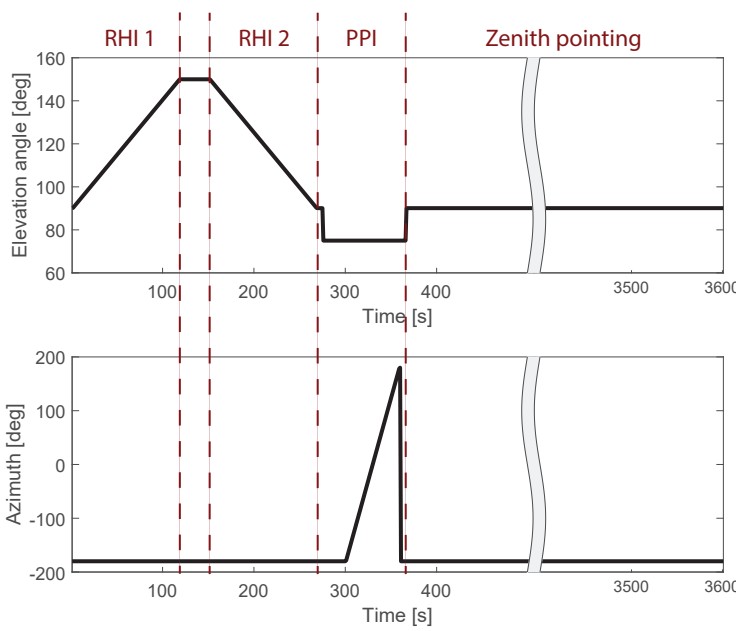

**Figure 1.** Temporal evolution of elevation angle (top) and azimuth angle (bottom) during the hourly scan cycle of SLDR Mira-35 as applied during CyCARE. Red-colored dashed vertical lines denote the time periods of the different RHI (range-height indicator), PPI (plan-position indicator) and zenith-pointing scan patterns.

to $150°$, corresponding to $30°$ elevation over the horizon toward north direction. This notation of the elevation angle range will be used throughout this article and figures.

Figure 1 describes the setup of one scan cycle as it was applied in the measurements of the SLDR mode MIRA-35, used in this study. Each scan cycle starts at minute 29 of each hour. Within 6.5 minutes, two RHI scans from $90°$ to $150°$ and from

$150°$ to $90°$ elevation angle and one Plan Position Indicator (PPI) scan at $75°$ elevation are performed. During the remaining 53.5 minutes of each measurement hour, vertical-stare observations (at $90°$ elevation angle) are performed to support standard retrievals, such as done within Cloudnet (Illingworth et al., 2007; Radenz et al., 2021) or as required for Doppler-spectra analysis techniques (Radenz et al., 2019; Bühl et al., 2019; Vogl et al., 2022; Schimmel et al., 2022). A limit of $150°$ elevation angle was established to avoid physical barriers like trees or buildings. It is also a reasonable compromise between required

horizontal homogeneity and the intensity of the SLDR gradient produced by the observed hydrometeors. As the detailed procedure of data acquisition was depicted by Görsdorf et al. (2015) and Myagkov et al. (2016b), the determination of the polarimetric parameters required for this study is only briefly outlined below. The primary measurement parameters are thus the Doppler power spectra received by the detectors in the co- and cross-channels with respect to the emitted polarization plane $P_{\mathrm{co}}(\omega_k)$ and $P_{\mathrm{cx}}(\omega_k)$, respectively, with $\omega_k$ being the Doppler frequency shift of each individual spectral component $k$. The

herein presented VDPS method only considers the main peak of the detected Doppler spectrum in the co-channel. Thus, in a

next step, each data point is screened for the spectral component $\omega_k^{\mathrm{max}}$ where $P_{\mathrm{co}}(\omega_k)$ is maximal. The following parameter is then only calculated for the Doppler spectral bin $\omega_k^{\mathrm{max}}$. The frequency dependency is thus omitted in the following and the polarimetric properties linear depolarization ratio in slanted mode (SLDR) can be derived as follows:

$$\mathrm{SLDR} = \frac{\langle \mathrm{P_{cx}} \rangle}{\langle \mathrm{P_{co}} \rangle} \tag{1}$$

Where $\langle \rangle$ denotes averaging over a number of collected Doppler spectra.

The raw spectra of SNR are subject to noise artifacts. Correspondingly, a noise filtering is performed to remove values which are below a given threshold value

$$n = m + 3\sigma \tag{2}$$

with $m$ being the mean and $\sigma$ being the standard deviation of noise in the co-channel. The properties of the noise in the co-
channel is estimated from the last 5 range gates of each profile assuming no scattering is present. A spectral line with the power in the cross channel below $n$ is excluded from the following analysis.

An important technical aspect which needs to be considered in the data analysis is the leakage of a fraction of signal from the co channel into the cross channel. The co-cross-channel isolation was determined with the experimental approach described in Myagkov et al. (2015), by means of identification of the minimum SLDR value that was measured at zenith-pointing, in the
presence of light drizzle. The co-cross-channel isolation used in this study was thus found to be $-35$ dB with MIRA-35.

## 2.2    Dataset

MIRA-35 is operated as part of the Leipzig Aerosol and Cloud Remote Observations System (LACROS, Radenz et al. (2021)), a suite of ground-based instruments of the Leibniz Institute for Tropospheric Research (TROPOS). Besides the SLDR-mode Ka-band scanning cloud radar, LACROS comprises an extensive set of active and passive remote sensing instruments for the
characterization of aerosol properties, clouds, and precipitation, including multi-wavelength polarization lidar, Doppler lidar, microwave radiometer and optical disdrometer. Data used in this study were acquired in the framework of a deployment of LACROS at the Mediterranean site of Limassol, Cyprus ($34.68°$ N, $33.04°$ E, 10 m a.s.l) during the Cyprus Clouds, Aerosol and pRecipitation Experiment (CyCARE, Ansmann et al. (2019); Radenz et al. (2021)). The region of Cyprus is a relevant location for studies of the impact of aerosol on cloud processes because of a large variety of air pollutants, desert dust, and
marine salt particles in the atmosphere above the island. The CyCARE campaign was conducted from September 2016 until March 2018 and aimed on the determination of the relationship between aerosol properties and the formation of cirrus and mixed-phase clouds (Ansmann et al., 2019; Radenz et al., 2021) in the heterogeneous freezing regime.

## 2.3    Measured SLDR and modeled $\widehat{\mathrm{SLDR}}$

The VDPS method combines simulations of $\widehat{\mathrm{SLDR}}$ (thereby and hereafter, the symbol $\hat{}$ denotes simulated parameters) with
measurements of SLDR (see Section 3). The study is based on the same set of equations as was previously presented by

Myagkov et al. (2016a). The theoretical framework assumes Rayleigh scattering and utilizes a spheroidal approximation of particle shape (Matrosov, 1991). In the used scattering model, polarimetric variables depend on two parameters: the polarizability ratio $\xi$, which describes the particles by means of a density-weighted axis ratio, and the degree of orientation $\kappa$ which is a measure of the preferred orientation of the spheroids population. It is well known that the Rayleigh approximation is not

always applicable to simulate scattering from individual and large ice particles at wavelengths shorter than C-band, which holds especially for absolute values such as reflectivity factor (Lu et al., 2016). At shorter wavelengths, the direct dipole approximation (DDA Draine and Flatau, 1994) can be used to simulate scattering of individual ice particles having a complex shape. Meanwhile, extensive databases exist (Lu et al., 2016) and found, e.g., special attention already for the application of multi-wavelength radar studies (von Terzi et al., 2022). However, these simulations and associated studies are often limited to

a number of predefined shapes and therefore do not necessarily represent the realistic distribution of ice particles observed by a radar (Leinonen et al., 2018). Simulations for a single particle also do not reflect the volumetric scattering effects of a large population of hydrometeors. In general, ice particles in a scattering volume have arbitrary shapes and the contribution of individual particles to the backscattering radar observables and especially polarimetric quantities is averaged out (Matrosov, 2021; von Terzi et al., 2022). We decided to assume the Rayleigh scattering and the spheroidal particle approximation (Matrosov,

1991; Ryzhkov, 2001; Bringi and Chandrasekar, 2001) because (1) such a model explains general polarimetric scattering effects with just a few parameters (axis ratio, permittivity and canting angle), (2) the model parameters are well constrained by the observations, (3) the volumetric scattering is taken into account, and (4) the model allows a computationally effective derivation of the polarizability ratio. In this study, we will sort particles into three primary categories based on their shape: oblate particles, which have a polarizability ratio less than one, prolate particles, characterized by a polarizability ratio greater

than one, and isometric particles, where the polarizability ratio is ranged from 0.8 to 1.2 , depending on the radar calibration (see Table 2). With respect to the definition in this study, we consider particles as isometric when they do not produce considerable polarimetric signatures. Such particles have either spherical or just slightly-non-spherical shape. In the case of particles with a low refractive index (i.e., low permittivity), their reduced response to radar waves may lead to scattering characteristics that resemble those of isometric particles.

Figure 2 shows dependencies of $\widehat{\text{SLDR}}$ on the polarizability ratio and degree of orientation of ice particles at $90°$ (zenith) and $150°$ ($60°$ off-zenith) elevation angle. Figures 2a and 2b show that $\widehat{\text{SLDR}}$ is mostly sensitive to $\xi$ (as noted by Matrosov et al. (2001)), which demonstrates the relevance of using SLDR rather than ZDR to determine the particle shape. For our radar configuration, the realistic range of possible polarizability ratios $\xi$ spans from 0.3 to 2.3 and the degree of orientation $\kappa$ is ranging from $-1$ to 1. $\kappa$ will only be briefly elaborated in this section as it will be used only qualitatively in the frame of

this study. In the case of spheroidal approximation and Rayleigh scattering regime, the polarizability ratio $\xi$, describing the shape of particles, is a function of permittivity and axis ratio and is independent of the particle volume. A polarizability ratio $\xi = 1$ designates spherical particles or particles with low density, while $\xi < 1$ and $\xi > 1$ describe oblate and prolate particles, respectively. Also for non-isometric particles, a decrease in apparent particle density causes $\xi$ to approach a value of unity (Myagkov et al., 2016a). The degree of orientation characterizes the width of the particle orientation angle distribution (the

degree of orientation is explained in more details in Myagkov et al. (2016a), in Figure 9 and Equation 11 in there). For instance,

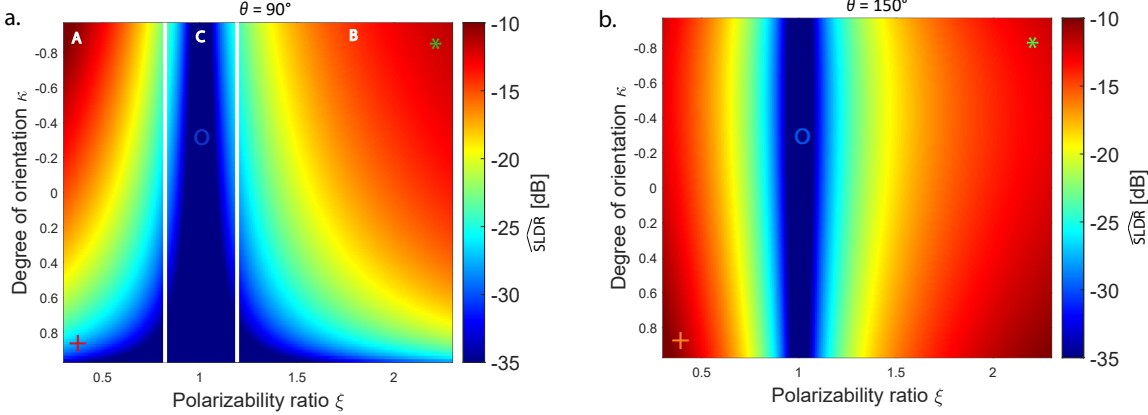

**Figure 2.** Modeled $\widehat{\mathrm{SLDR}}$ as function of polarizability ratio $\xi$ and degree of orientation $\kappa$ for particles at (a) $\theta = 90°$ and (b) $\theta = 150°$ antenna elevation angle. '+' (oblate particles), 'O' (isometric particles) and '*' (prolate particles) symbols are data points used in Figure 3. The elevation dependency of these three scenarios are depicted further in Section 2.3. The two white vertical lines in (a) separate the three particle domains of oblate (zone A), prolate (zone B) and the isometric (zone C) hydrometeors.

$|\kappa| = 0$ corresponds to uniform distribution, while $|\kappa| = 1$ indicates that all particles are aligned in the same way. The sign of $\kappa$ indicates the preferable orientation of the symmetry axis, i.e., $\kappa = +1$ indicates that all particles are aligned and have a vertical symmetry axis, $\kappa = -1$ corresponds to the case when particles have a predominantly horizontally aligned symmetry axis. We therefore assume $\kappa >= 0$ for oblate particles and $\kappa <= 0$ for prolate particles. Regarding Figures 2a and 2b we consider that

$\kappa \approx -0.3$ is corresponding to randomly oriented isometric particles when SLDR is minimal and these values do not depend on the elevation angle (Myagkov et al., 2016a).

     A subset from Figure 2 is presented in Figure 3 in order to demonstrate the general, idealized relationship between $\widehat{\mathrm{SLDR}}$ and elevation angle for the main particle shape classes oblate ("+"), isometric ("o") and prolate ("-"), thereby assuming predominantly horizontal orientation. Indeed, the "+" symbol is located in the oblate domain (zone A) described by a polarizability

ratio $\xi = 0.35$ and a degree of orientation $\kappa = 0.85$ representing horizontally oriented plate-like particles, while the "*" symbol is located in the prolate domain (zone B) described by $\xi = 2.15$ and $\kappa = -0.85$ representing horizontally oriented columnar crystals. The Symbol "o" is determined by $\xi = 1$ and $\kappa = -0.4$, such as randomly oriented spherical particles like liquid droplets (Myagkov et al., 2016a), which is representative for the isometric domain (zone C). A value of $\widehat{\mathrm{SLDR}}$ is derived for all elevation angles from $90°$ and $150°$, leading to Figure 3 which links our study to findings of Matrosov et al. (2012) showing

distinct elevation-dependent signatures of SLDR for particles with different shapes. As illustrated in Figure 3, prolate particles are characterized by nearly constant and relatively high values of SLDR at all elevation angles, which reach values of around $-25$ dB for solid columns and more than $-20$ dB for pronounced needles of high axis ratio (Reinking et al., 2002; Matrosov et al., 2012). The isometric primary particle shape class is represented by constantly low values of SLDR at all elevation angles between $90°$ to $150°$. Finally, plate-like particles, belonging to the oblate particle class, known to align predominantly hori-

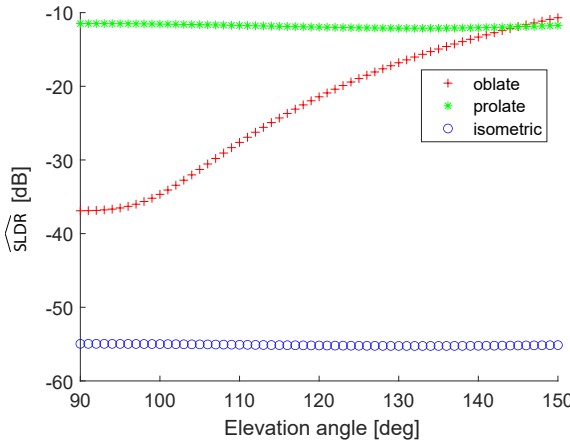

**Figure 3.** Distributions of modeled $\widehat{\mathrm{SLDR}}$ as a function of elevation angle between $90°$ and $150°$ for typical horizontally oriented oblate ("+" red), isometric ("o" blue), and prolate ("*" green) particles, respectively. The same symbols in Figure 2 illustrate the location of the data points in the model field at $90°$ (Figure 2a) and $150°$ (Figure 2b) elevation angle.

zontally along their planar planes, produce scattering similar to isometric particles observed at zenith-pointing ($90°$ elevation angle) and will increasingly appear oblate at low elevation angles. That is why in the case of plate-like hydrometeors, $\mathrm{SLDR}$, representative of the particle shape, is minimal at zenith-pointing and increases from $90°$ to $150°$ elevation angle. Indeed, at zenith-pointing, plate-like crystals have random orientation in the polarization plane, while at a low elevation angle horizontally aligned particles produce rather coherent returns in both polarimetric channels. Note, that it is not directly possible to classify

the type of isometric particles (e.g., aggregates or rimed particles can be isometric particles, too) since they have similar angular polarimetric signatures at all elevation angles. Discrimination between these types of particles can be done, e.g., using multiple-frequency observations (Kneifel et al., 2016) but this is out of the scope of the current study. VDPS method aims to differentiate the three main particle shape classes and their vertical evolution within cloud systems in order to determine microphysical processes occurring in mixed-phase clouds.

**3   Methodology**

The concept of the VDPS approach is to realize a tailored retrieval of the vertical distribution of particle shape. The VDPS method, adapted for the SLDR-mode scanning cloud radar as introduced in Section 2.1, has the particularity of combining simulated and measured values of SLDR at only two elevation angles, isolated from a full RHI scan. As the VDPS method relies on polarimetric measurements at different elevation angles, horizontal homogeneity of the observed clouds is required.

The scale of the horizontal homogeneity is defined by the maximum observation distance of the used cloud radar and the lowest elevation angle (10–15 km and $30°$, respectively). Thus, the required scale of the horizontal homogeneity is mostly

below 13 km, which is comparable, e.g., to a footprint of a space-borne passive microwave sensor. A majority of stratiform clouds have much larger spatial scale. In addition, the algorithm requires a minimum number of data points in each layer, representing $15\%$ of the total amount of data points, as will be explained in Section 3.1.

The general flow chart describing the three-step procedure is depicted in Figure 4. Within the first step, presented in Section 3.1, the dataset is prepared for the evaluation against the spheroidal scattering model in Section 3.2. By combination of $\widehat{\mathrm{SLDR}}$ simulated by the spheroidal scattering model with SLDR observations, the range of possible primary particle shape classes is identified and the associated uncertainties are assessed in Section 3.2. In the final step presented in Section 3.3, linear regressions of SLDR vs. elevation angle are calculated and deployed to identify the correct primary particle shape class and to

assign the proper polarizability ratio $\xi$ from the set of possible solutions determined in Section 3.2.

### 3.1   Determination of SLDR at the boundaries of the elevation range

For each of the four individual scan patterns described in Figure 1, the returned signals in the co- and cross-channel $P_{\mathrm{co}}(\omega_k)$ and $P_{\mathrm{cx}}(\omega_k)$, respectively, collected by MIRA-35 are saved in a level-0 file, in the pdm format defined by Metek company. Consequently, the pdm data are in a first step converted into NetCDF format containing the polarimetric measurements of

$\mathrm{SLDR}(\omega_k)$, calculated with Equation (1), as well as elevation angle and range. Next, the noise filtering (equation (2)) is applied as explained in Section 2.3 and only the maximum spectral component of the remaining noise-free spectra are selected. Thus, arrays containing one value of SLDR per elevation angle are obtained for each granule of time and range. All range values are converted into height above ground, using the elevation angle $\theta$ as additional input. The VDPS algorithm runs automatically for each selected RHI scan. A main loop is used to separate the observations into multiple vertical 'height'

layers. In general, any arbitrary value of height resolution can be chosen. For the current study, each height step corresponds to the range resolution of MIRA-35 (31,18 m, i.e., the height resolution at zenith-pointing), similar to as was done by Myagkov et al. (2016a). The following procedure is performed for each height layer which contains at least 20 values of SLDR from a full RHI scan recorded from $90°$ to $150°$ elevation angle (Figure 1). The value of 20 points per layer represents about $15\%$ of the maximal number of data points. If this limit is not reached, it could mean that no cloud was detected at this layer or that not enough particles are contained at the investigated height level of the cloud, which would influence the quality of results. In

this situation, the procedure will be stopped only for this layer at this step (no results are produced) and will continue to iterate into the next layer. If a sufficient amount of data points was found at a height level, a new vector of $\mathrm{SLDR}(\mathrm{H},\theta)$ is built. The elevation range of $\mathrm{SLDR}(\mathrm{H},\theta)$ does thus not necessarily span the full elevation range of the RHI scan, as some data points at the elevation limits might have been removed.

As shown in Figure 3 and as will be elaborated further in Section 3.3, polarimetric signatures of different particle shapes are most visible when the elevation angle difference of the performed scans is large. For this reason, a full RHI scan is used to verify the homogeneity of the investigated cloud (Section 3.1) and to calculate the SLDR linear regression (Section 3.3), but only values of SLDR at two elevation angles are needed in the model output (Section 3.2). In order to prepare the observational input for the evaluation against the spheroidal scattering model to be described in Section 3.2, we will look for

the data points of SLDR associated to the smallest observed value of elevation angle ($\theta_{\mathrm{min}}$, usually zenith-pointing) and to

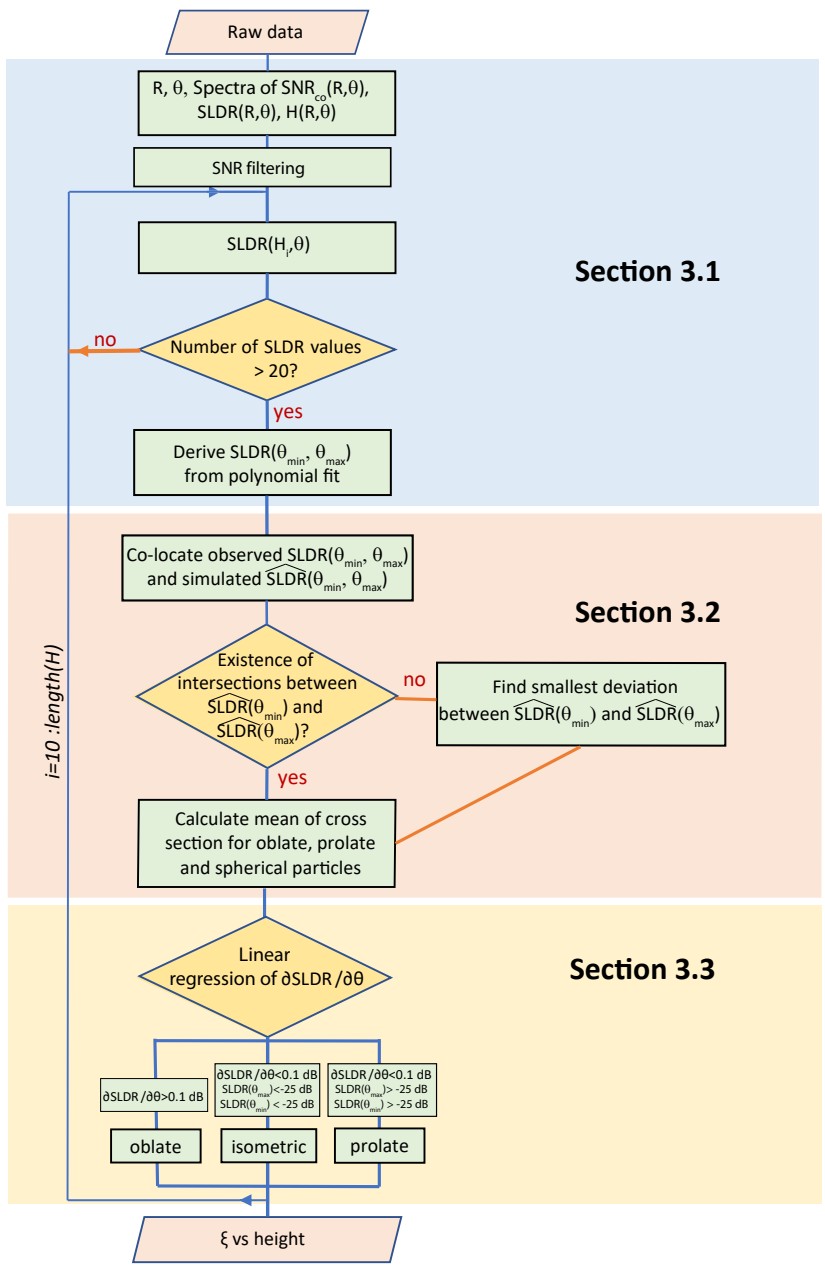

**Figure 4.** Flowchart describing the VDPS method.

the largest value of elevation angle ($\theta_{\max}$, usually $150°$). Thus, in a next step, fit values of measured SLDR at the minimum elevation angle $\theta_{\min}$ (SLDR($\theta_{\min}$)) and at maximum elevation angle $\theta_{\max}$ (SLDR($\theta_{\max}$)) are calculated. These notations will be used further in Section 3.2. It can be seen in Figure 3, that the relationship between SLDR and elevation angle is not linear for SLDR, especially in the case of oblate particles, and the more appropriate method to calculate SLDR($\theta_{\min}$) and

SLDR($\theta_{\max}$) for all cases is to use a $3^{\text{rd}}$ degree polynomial fit. SLDR($\theta_{\min}$) and SLDR($\theta_{\max}$) are determined with the fit values from the $3^{\text{rd}}$ degrees polynomial fit at $\theta_{\min}$ and $\theta_{\max}$, respectively. As an example, Figure 5 shows the distribution of SLDR from $\theta_{\min}$ to $\theta_{\max}$. Values of SLDR($\theta_{\min}$) and SLDR($\theta_{\max}$) are readable at $90°$ ($\theta_{\min}$) and $150°$ ($\theta_{\max}$) elevation angle as SLDR($\theta_{\min}$) $= -32$ dB and SLDR($\theta_{\max}$) $= -11$ dB. SLDR($\theta_{\min}$) and SLDR($\theta_{\max}$) are saved and will be utilized in Section 3.2 for the evaluation against the spheroidal scattering model compiled at the same elevation angles $\theta_{\min}$ and $\theta_{\max}$.

## 3.2 Estimation of the polarizability ratio for each layer

In the first step of the VDPS retrieval we find two SLDR values corresponding to $\theta_{\min}$ and $\theta_{\max}$ (Section 3.1). In the second step we search for values of the polarizability ratio and the degree of orientation for which the simulated $\widehat{\text{SLDR}}$ fits to SLDR($\theta_{\min}$) and SLDR($\theta_{\max}$).

The original spheroidal scattering model based on Myagkov et al. (2016a) does not take into account hardware-related

effects and, therefore, predicts minimum values of $\widehat{\text{SLDR}}$ that cannot be reached with the current radar technology due to the polarimetric coupling in the antenna system. The polarimetric coupling (co-cross channel isolation) of the used radar is $-35$ dB, as mentioned in Section 2.1, and leads to an increased uncertainty of the retrieval for particles with polarizability ratio between 0.9 and 1.1. The modeled distribution of $\widehat{\text{SLDR}}$ from $90°$ to $150°$ elevation angle for three exemplary particle habits oblate, isometric, and prolate are illustrated in Figure 3. This graphic represents the theoretical relationship between $\widehat{\text{SLDR}}$ and

elevation angle in the three different primary particle shape classes, which is about to be faced with the direct measurements of SLDR. In the second part of this section, we compare the modeled $\widehat{\text{SLDR}}$ and measured SLDR obtained from the polynomial fit (Section 3.1) at elevation angles $\theta_{\min}$ and $\theta_{\max}$, as explained in Section 3.1. In order to consider potential measurement inaccuracies, the $95\%$ confidence interval $\Delta_{95}$ of the polynomial fit will be used to determine the potential range of the intersection. The confidence interval is calculated as follows:

$$\Delta_{95} = 2\Delta \tag{3}$$

where $\Delta$ is the standard deviation of the difference between the measured and simulated values of SLDR at all available elevation angles from $90°$ to $150°$. The model is processed at $\theta_{\min}$ and $\theta_{\max}$ and the algorithm identifies isolines of SLDR($\theta_{\min}$) $= \widehat{\text{SLDR}}(\theta_{\min}) \pm \Delta_{95}$, and SLDR($\theta_{\max}$) $= \widehat{\text{SLDR}}(\theta_{\max}) \pm \Delta_{95}$, in the modeled fields of $\widehat{\text{SLDR}}$ at $\theta_{min}$ and $\theta_{max}$, respectively. For example, in Figure 6a and 6b we can see the isoline where SLDR($\theta_{\min}$) $= \widehat{\text{SLDR}}(\theta_{\min})$ plotted in

red and the isoline where SLDR($\theta_{\max}$) $= \widehat{\text{SLDR}}(\theta_{\max})$ plotted in blue on the model, respectively. The two isolines are plotted together in Figure 6c, highlighting intersections between $\widehat{\text{SLDR}}(\theta_{\min})$, shown as red curve, and $\widehat{\text{SLDR}}(\theta_{\max})$, shown as blue curve, resulting in $\xi = 0.45$ and $\xi = 2$. If no intersection is found between $\widehat{\text{SLDR}}(\theta_{\min})$ and $\widehat{\text{SLDR}}(\theta_{\max})$, the algorithm searches for the point where the difference between $\widehat{\text{SLDR}}(\theta_{\min})$ and $\widehat{\text{SLDR}}(\theta_{\max})$ is the lowest. Finally, the algorithm char-

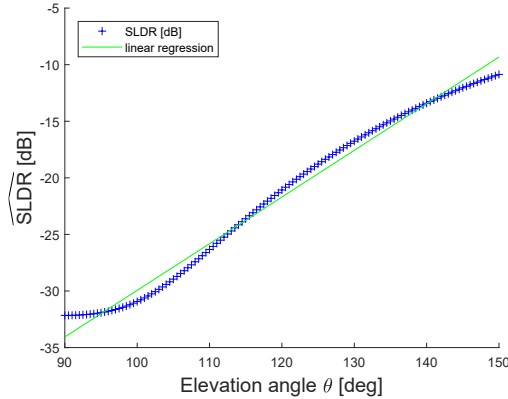

**Figure 5.** Distribution of SLDR as a function of elevation angle between $\theta_{\min} = 90°$ and $\theta_{\max} = 150°$ for the same dendritic crystal population as presented in Figure 6 : $\mathrm{SLDR}(\theta_{\min}) = -32\mathrm{dB}$, $\mathrm{SLDR}(\theta_{\max}) = -11\mathrm{dB}$, and $\kappa = 0.85$. The green line represents the SLDR linear regression calculated in Section 3.3.

acterizes the x-axis positions (polarizability ratio $\xi$) by deriving the mean and standard deviation of all overlapping data points
included in each intersection between the isolines of $\widehat{\mathrm{SLDR}}(\theta_{\min})$ and $\widehat{\mathrm{SLDR}}(\theta_{\max})$ (Figure 6). Three values of $\xi$ are saved at each height iteration corresponding to the three primary particle shape classes : the first intersection in the oblate particle shape class with $\xi < 1$ ($\xi = 0.45$ in Figure 6c), the second intersection for the prolate particle shape class with $\xi > 1$ ($\xi = 2$ in Figure 6c) and a mean of these two intersections for the isometric or low-density particle shape class with $\xi \approx 1$. The procedure could be repeated in a similar manner for determination of the possible y-axis values, which are the possible solutions of the degree
of orientation $\kappa$, which is however not in the scope of our study.

### 3.3   Identification and quantification of the primary particle shape class

The last step of the VDPS method consists of the identification of the primary particle shape class among the three possible solutions introduced in Section 3.2 and to quantify the primary particle shape class with the assigned value of $\xi$. As introduced in Section 2.3, the relationship between SLDR and the elevation angle is an important aspect to determine the particle shape
(Reinking et al., 2002; Matrosov et al., 2005, 2012) and will be used in the following to discriminate between the primary particle shape classes. A threshold of $\frac{\partial \mathrm{SLDR}}{\partial \theta}$ is determined in such a way that an unambiguous separation of the prolate, oblate and isometric hydrometeor shape classes is possible, by applying a robust linear fit to all observed pairs of SLDR and elevation angle. The resulting limit values were derived to be $lim_{\mathrm{SLDR}} = 0.1$ dB, as a threshold describing a certain change of the SLDR in dB per degree of elevation angle, and $lim_{\mathrm{pro}} = -25$ dB, which describes the maximum value of SLDR to be associated
to the prolate shape class. It should be noted that the two limit values might depend on the individual radar calibration. The actual shape class selection criteria are summarized in Table 2 and are described in the following. If the linear regression $\frac{\partial \mathrm{SLDR}}{\partial \theta}$ exceeds $lim_{\mathrm{SLDR}}$, particles are assigned to the oblate primary particle shape class. If $\frac{\partial \mathrm{SLDR}}{\partial \theta}$ doesn't exceed $lim_{\mathrm{SLDR}}$

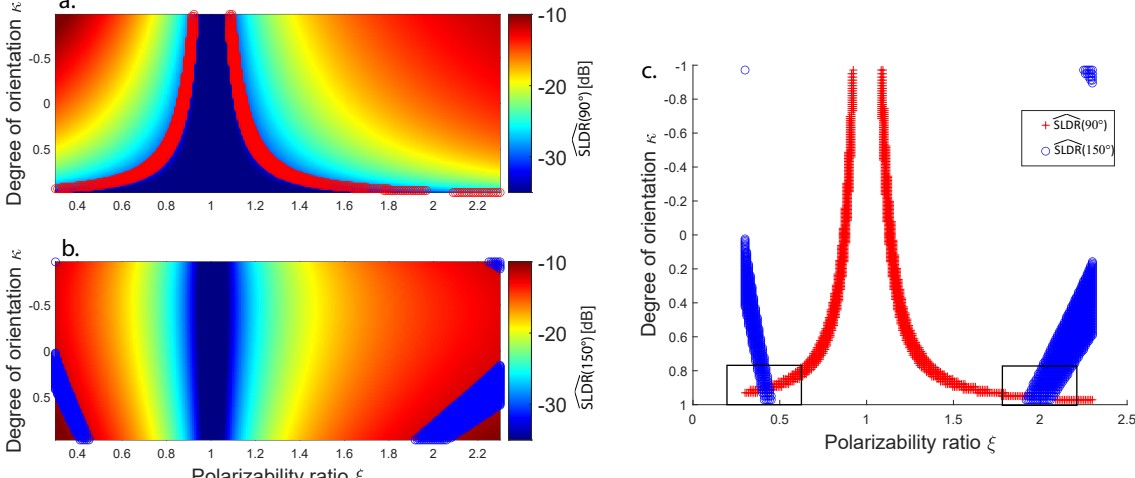

**Figure 6.** Determination of the possible values of $\xi$ by searching for the intersections between $\widehat{\mathrm{SLDR}}(\theta_{\min})$ and $\widehat{\mathrm{SLDR}}(\theta_{\max})$ on the spheroidal scattering model at (a) $\theta_{\min} = 90°$ and (b) $\theta_{\max} = 150°$ elevation angle. The red and blue curves in (a), (b) and (c) depict the isolines as (a) $\mathrm{SLDR}(\theta_{\min}) = \widehat{\mathrm{SLDR}}(\theta_{\min})$ and (b) $\mathrm{SLDR}(\theta_{\max}) = \widehat{\mathrm{SLDR}}(\theta_{\max})$ at $90°$ and $150°$ elevation, respectively. In (c) the intersections of the $\widehat{\mathrm{SLDR}}(\theta_{\min})$ and $\widehat{\mathrm{SLDR}}(\theta_{\max})$ isolines are shown. As input, hypothetical values of typical oblate particles with $\mathrm{SLDR}(\theta_{\min}) = -32\mathrm{dB}$ and $\mathrm{SLDR}(\theta_{\max}) = -11\mathrm{dB}$ were selected.

as well as $\mathrm{SLDR}(\theta_{\min})$ and $\mathrm{SLDR}(\theta_{\max})$ exceed $lim_{\mathrm{pro}}$, particles are assigned to the prolate primary particle shape class. If $\frac{\partial \mathrm{SLDR}}{\partial \theta}$ doesn't exceed $lim_{\mathrm{SLDR}}$ as well as $\mathrm{SLDR}(\theta_{\min})$ and $\mathrm{SLDR}(\theta_{\max})$ are below $lim_{\mathrm{pro}}$, particles are associated to the

isometric primary particle shape class. If particles are assigned to the isometric particle shape class, $\xi$ will be calculated as the mean of the associated values of $\xi$ contained in both intersections of $\widehat{\mathrm{SLDR}}(\theta_{\min})$ and $\widehat{\mathrm{SLDR}}(\theta_{\max})$ on both sides of $\xi = 1$ (Section 3.2). In the oblate and prolate primary particle shape classes, the error bars are calculated based on the intersections of the standard deviation obtained for $\widehat{\mathrm{SLDR}}(\theta_{\min})$ and $\widehat{\mathrm{SLDR}}(\theta_{\max})$, following the same procedure as explained in Section 3.2. Concerning the isometric primary particle shape class, $\xi$ values of the two intersections identified before are used as error

bars. Figure 5 depicts the relationship between SLDR and elevation angle from $\theta_{\min}$ to $\theta_{\max}$. According to Reinking et al. (2002) and Matrosov et al. (2012), the found relationship is representative for oblate particles such as as plate-like crystals, as depicted in Table 2. Regarding Figure 6c presented in Section 3.2, we observe two intersections on both sides of $\xi = 1$ and the choice of one of them requires an evaluation of the linear regression of SLDR from $\theta_{min}$ to $\theta_{max}$. The associated distribution of SLDR presented in Figure 5 confirms the assignment of ice particles to the oblate primary particle shape class due to the

increase of SLDR from $\theta_{\min}$ to $\theta_{\max}$ and the exceeding of $lim_{\mathrm{SLDR}}$. A value of $\xi = 0.45$ is finally derived for this layer. The last step, according to the flow chart depicted in Figure 4, is to apply the classification to the previously calculated profile of $\xi$ (see Section 3.2) and to store the selected values. This distribution of particle shape delivers information about the vertical profile of ice particle shapes in a cloud which is a relevant indicator to understand in-cloud processes, illustrated in Section

4.4. The next section aims to evaluate and validate the VDPS method by means of three case studies, representing the three
previously described particle shape classes prolate, oblate, and isometric, and demonstrate the ability of the VDPS method to detect microphysical processes.

## 4 Results

In this Section, we will demonstrate the capabilities of the VDPS retrieval by means of three case studies associated with the three main particle shape classes isometric (rain, Section 4.1), prolate (columnar ice crystals, Section 4.2) and oblate (plate-like
ice crystals, Section 4.3). A fourth case study is presented in Section 4.4 to conclude and open the discussion concerning the ability of VDPS to describe microphysical processes by a change in particle shape from cloud top to cloud base. The four case studies were selected from the CyCARE observations, presented in Section 2. Temperature provides an important constraint for the particle shape, since laboratory studies show a clear relationship between particle shape, temperature and supersaturation with respect to ice (Bailey and Hallett, 2009; Myagkov et al., 2016b). Given conditions of liquid water saturation, near $T = $
$-2°$C, the growth is plate-like, near $T = -5°$C the growth is columnar, near $T = -15°$C the growth again becomes plate-like and at lower temperature, the growth becomes a mixture of thick plates and columns. A general meteorological situation is presented for each case study using the Cloudnet classification of targets based on MIRA-35 at zenith-pointing and auxilliary instrumentation (Illingworth et al., 2007) and a RHI scan of SLDR from $\theta_{\min}$ to $\theta_{\max}$. Subsequently, the polarimetric parameter SLDR measured at $\theta_{\min}$ and $\theta_{\max}$ is combined with the spheroidal scattering model introduced in Section 3.2. We will focus
only on the selected layer to illustrate the case studies even though all layers are processed to obtain the vertical distribution of particle shape. The last step aims to deliver insights into the quantification of the primary particle shape classes, as explained in Section 3.3, with the vertical distribution of $\xi$ in the investigated cloud. Since the proposed method uses the spheroidal approximation of pure-ice particles and assumes Rayleigh scattering, the derived values of $\xi$ should be analyzed with care when the method is applied to rain and close to the melting layer. Since rain droplets corresponding to the maximum spectral
line are often near spherical, $\xi$ is valid since for spherical particles it is not sensitive to the refractive index. In contrast, $\xi$ in the melting layer is likely not valid, because the depolarization observed in the melting layer is not caused by columnar shapes of particles but by particle's strongly irregular shapes, water coating (and associated fluctuations of apparent density), and their

**Table 2.** Assignment of the characteristic values of SLDR at $\theta_{\min} = 90°$ and $\theta_{\max} = 150°$ elevation angle and their linear regressions as function of $\theta$. The associated typical ranges of $\xi$ are given, as well. Please note, values of SLDR($\theta_{\min}$) and SLDR($\theta_{\max}$) for the isometric shape class correspond to the detection limit of SLDR (See Section 2.1). The limit values are $lim_{\mathrm{SLDR}} = 0.1$ dB and $lim_{\mathrm{pro}} = -25$ dB.

| Shape class | Linear regression | Value at $90°$ | Value at $150°$ | Polarizability ratio $\xi$ |
|:---:|:---:|:---:|:---:|:---:|
| Oblate | $\frac{\partial \mathrm{SLDR}}{\partial \theta} > lim_{\mathrm{SLDR}}$ | SLDR($\theta_{\min}$) = -30 dB | SLDR($\theta_{\max}$) = -10 dB | $\xi$ = 0.2 - 0.8 |
| Isometric | $\frac{\partial \mathrm{SLDR}}{\partial \theta} < lim_{\mathrm{SLDR}}$ | SLDR($\theta_{\max}$) = -35 dB | SLDR($\theta_{\min}$) = -35 dB | $\xi$ = 0.8 - 1.2 |
| Prolate | $\frac{\partial \mathrm{SLDR}}{\partial \theta} < lim_{\mathrm{SLDR}}$ | SLDR($\theta_{\min}$) = -20 dB | SLDR($\theta_{\max}$) = -20 dB | $\xi$ = 1.2 - 2.4 |

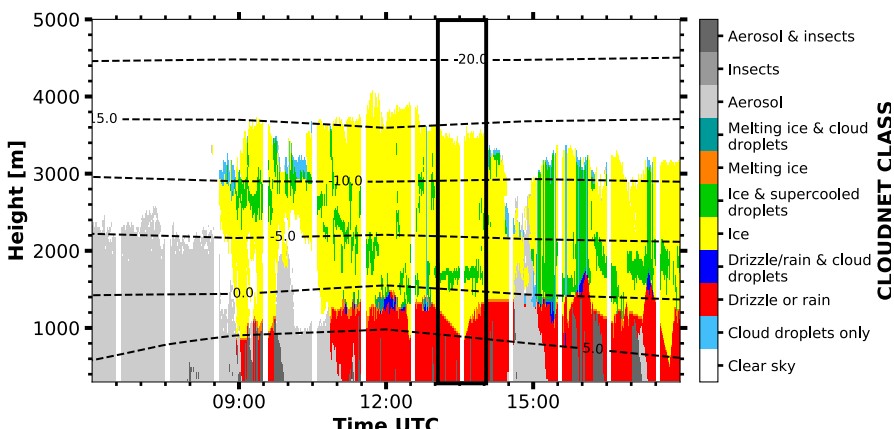

**Figure 7.** Cloudnet target classification mask as derived from observations at Limassol on 13 February 2017 from 06:00 to 18:00 UTC. The black box denotes the RHI scan that is discussed in further detail in Section 4.1.

large size. This section aims to demonstrate that the VDPS method gives concordant results with the observations for the three primary particle shape classes, isometric, prolate and oblate particles, introduced in Section 3.2, and that it is a promising
supplemental technique for studying cloud microphysical processes.

### 4.1 Isometric particle shape class: Rain event on 13 February 2017 at 13:31 UTC

The first case study concentrates on the occurrence of rain, i.e., hydrometeors representative for the primary isometric particle shape class. Measurements were recorded on 13 February 2017 during an RHI scan from 13:31 to 13:33 UTC at Limassol. The studied cloud system, enframed by the the black box in the Cloudnet target classification mask shown in Figure 7, was
identified to contain rain droplets at heights between 300 m and 1300 m. The sudden drop of the melting layer height from 1300 m to around 1000 m height that is visible right at the time of the RHI scan, is an artifact of the melting layer detection scheme of Cloudnet, which switched from a fall-velocity-based detection to the 0°C-dewpoint level as threshold for the melting layer identification. However, the actual melting layer is well recognizable in Figure 8 by means of the observed high values of SLDR at around 1300 m height. The Cloudnet classification indicates a mixed-phase layer at 1800 m height. For this case
study, we are particularly interested in the rain from 300 m to 1200 m height.

Figure 8 shows the RHI scan of SLDR from $90°$ to $150°$ elevation angle which were performed at 13:31 UTC. Values of SLDR from $\theta_{\min}$ to $\theta_{\max}$ are low (around $-30$ dB) and constant at heights below the melting layer, which is in agreement to what can be expected from scattering by isometric particles, as explained in Section 3.3. To illustrate this case study, we will focus only on one layer located at the height level from 868 m to 899 m, represented by the black line on the y-
axis in Figure 8. In Figure 9b, the intersection of $\widehat{\text{SLDR}}(\theta_{\min})$ and $\widehat{\text{SLDR}}(\theta_{\max})$ is detectable by the red and blue curves which match the data of SLDR at $\theta_{\min}$ and $\theta_{\max}$, respectively, with the simulated data $\widehat{\text{SLDR}}$ from the spheroidal scattering

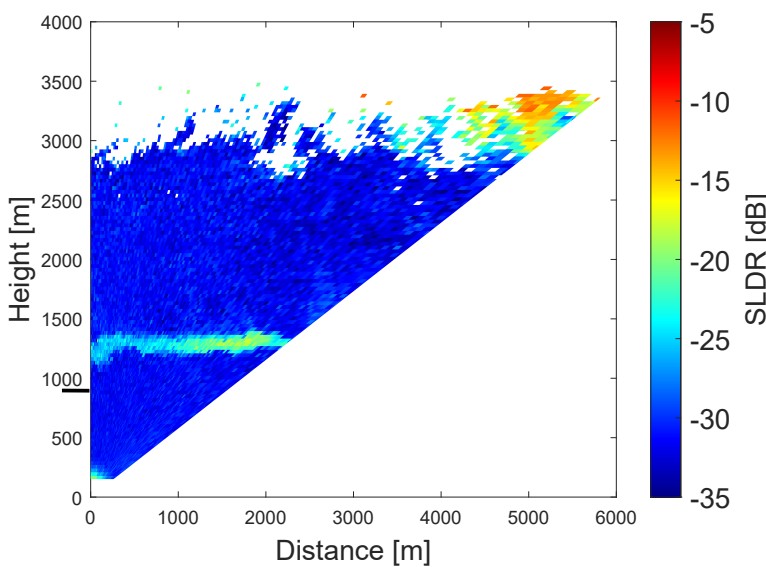

**Figure 8.** RHI scan of SLDR observed on 13 February 2017, at 13:31 UTC in Limassol from $90°$ to $150°$ elevation angle. The black horizontal line on the y axis mark the height of the layer analysed in Figure 9.

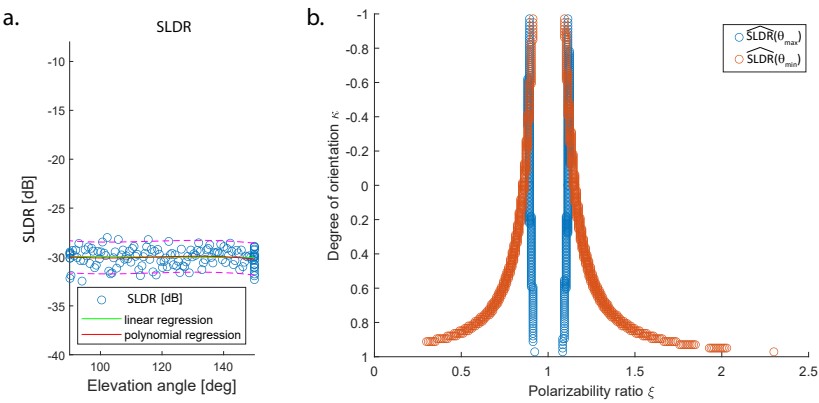

**Figure 9.** Detailed view into the isometric-shape case study presented in Figure 8 for the layer from 868 m to 899 m height. (a) Distribution of measured values of SLDR from $\theta_{min}$ to $\theta_{max}$ elevation angle and associated linear and polynomial fits. The dashed pink lines in (a) correspond to the 95% prediction interval from the third degrees polynomial function, used to determine the intersection of $\widehat{SLDR}(\theta_{min})$ and $\widehat{SLDR}(\theta_{max})$. (b) Intersection between $\widehat{SLDR}(\theta_{min})$ and $\widehat{SLDR}(\theta_{max})$ at $\theta_{min}$ and $\theta_{max}$, respectively.

model. We can distinctly notice the presence of two intersections between $\widehat{\mathrm{SLDR}}(\theta_{\min})$ and $\widehat{\mathrm{SLDR}}(\theta_{\max})$ from either side of the dashed red line, resulting in $\xi = 0.9$ and $\xi = 1.1$. In Figure 9a, the slope of the linear regression $\frac{\partial \mathrm{SLDR}}{\partial \theta}$ is constant between $\theta_{\min}$ and $\theta_{\max}$ where $\mathrm{SLDR}(\theta_{\min}) < \lim_{\mathrm{pro}}$ and $\mathrm{SLDR}(\theta_{\max}) < \lim_{\mathrm{pro}}$ and $\frac{\partial \mathrm{SLDR}}{\partial \theta} < lim_{\mathrm{SLDR}}$. Regarding Table 2, this configuration describes the isometric primary particle shape class. Finally, the vertical distribution of $\xi$ in the cloud is calculated following Section 3.3 and is shown in Figure 10. Concerning the observations, the melting layer is well identified by a variable $\xi$, as explained in introduction of Section 4, in the height range from 1250 m to 1350 m. Below this layer, $\xi$ takes values around 1 which describes isometric or less dense particles (Section 3.2). Looking at the Cloudnet classification (Figure 7), the drizzle-or-rain class dominates the measurement at heights below approximately 1000 m height, which can be extended to the melting layer at around 1300 m height, taking in account the misidentified drop due to the melting layer detection of Cloudnet, as previously explained. Figure 7 shows, in the black box, a temperature higher than $0°\mathrm{C}$ in this layer, which confirms the presence of liquid droplets, i.e., isometric particles. Application of the VDPS approach results in derivation of the same isometric primary particle shape class as determined based on the auxiliary observations (temperature and Cloudnet classification). With respect to the presented case it is noteworthy that it is likely that the observed rain droplets were small in size. This is corroborated by the absence of any elevation dependency of SLDR (Figure 9). In the case of strong rain, the oblateness of droplets would become apparent as SLDR increases from zenith pointing to $150°$ elevation angle, as we observed in some situations of convective rain at Limassol during the CyCARE campaign (Section 2.2). Above the melting layer from 1700 m to 2800 m height, the VDPS method derived isometric or less dense particles, as well. Given that temperatures are below freezing level at these heights and that Cloudnet identified a mix of ice and supercooled droplets, it is likely that these isometric or less dense particles are the result of mixed-phase cloud processes, such as riming or aggregation, which cannot unambiguously be identified solely with the VDPS method. Based on the VDPS method, the height level of the particle shape transition can be determined to be present at around 2800 m. Above, $\xi$ was found to be well below 1, representing oblate particles, whose formation is also corroborated by the ambient temperatures of around $-15°\mathrm{C}$ at this height level (see Figure 7). Applicability of the VDPS method is in the present case limited with respect to the interpretation of the microphysical process which led to the formation of the layer with isometric particle shape between approximately 1500 m and 2700 m height. Doppler spectral methods or multi-frequency approaches could help here to investigate the possible contributions of riming and aggregation (Kneifel et al., 2016; Radenz et al., 2019; Kalesse-Los et al., 2022; Vogl et al., 2022).

### 4.2 Prolate particle shape class: Columnar crystals on 4 January 2017 at 04:30 UTC

The second case study chosen to evaluate the VDPS method is dedicated to the characterization of columnar crystals. The corresponding measurement was recorded in Limassol on 8 December 2016 during an RHI scan from 00:31 to 00:33 UTC. Figure 11 presents the Cloudnet classification for the time range from 00:00 to 03:00 UTC on 8 December 2016, with the selected case study marked by the black frame. Figure 12 shows the RHI scans of SLDR from $90°$ to $150°$ elevation angle at 00:31 UTC. In this RHI scan, high values of SLDR are observed at all elevation angles (between $-20$ dB and $-15$ dB), suggesting that the cloud is well homogeneous and that ice particles have a high capability to depolarize the returned radar signals. According to Reinking et al. (2002), particles having a SLDR from $-20$ dB to $-15$ dB can be classified at first glance

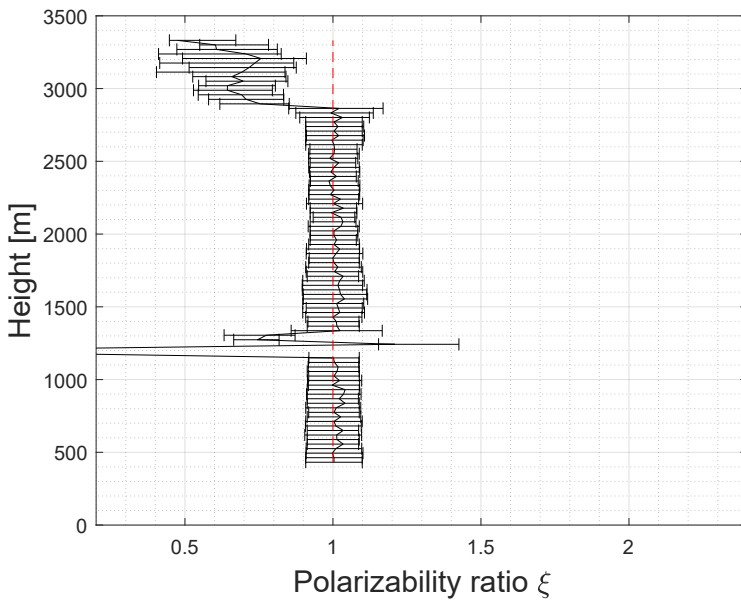

**Figure 10.** Vertical distribution of $\xi$ as calculated with the VDPS method for each layer of the isometric-shape case study observed in Limassol on 13 February 2017, 13:31 UTC.

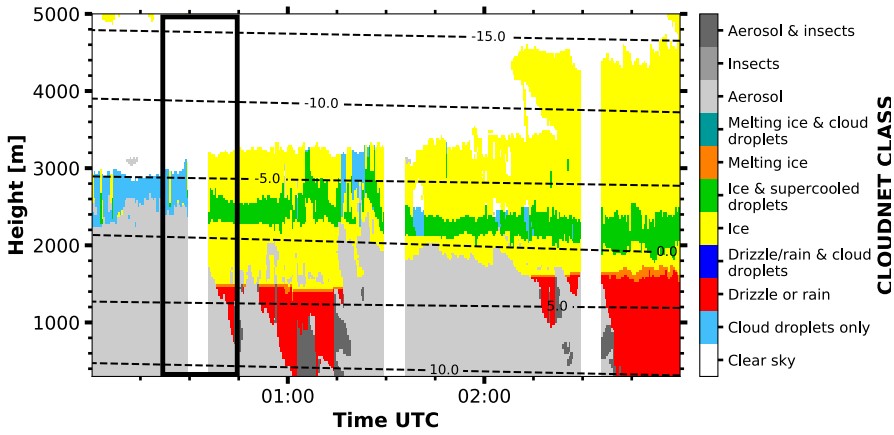

**Figure 11.** Cloudnet target classification mask as derived for observations at Limassol on 8 December 2016 from 00:00 to 03:00 UTC. The black box denotes the RHI scan that is discussed in further detail in Section 4.2.

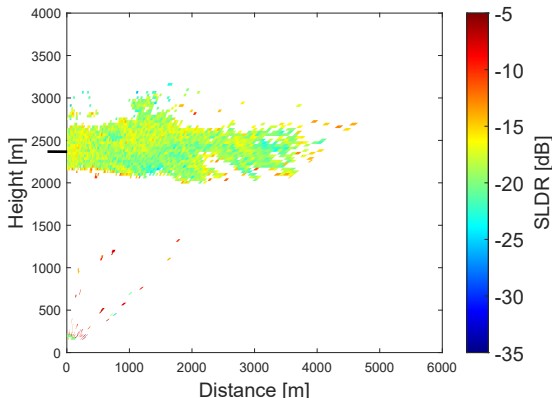

**Figure 12.** RHI scan of SLDR on 8 December 2016, at 00:31 UTC, Limassol, from $90°$ to $150°$ elevation angle. The black horizontal line on the y axis mark the height of the layer analysed in Figure 13.

as needles or hollow columns. This constellation excludes isometric particles and oblate particles and is a specific property of columnar crystals (see Table 2). As for the first case study, the retrieval is visualized only for one specific layer, which in this case spans from 2458 m to 2490 m height, indicated by the black line on the y-axis in Figure 12. Figure 13a shows the SLDR linear regression represented by the green line, which confirms that $\frac{\partial \text{SLDR}}{\partial \theta} < lim_{\text{SLDR}}$. The polynomial fit represented by the red curve is used at $\theta_{\min}$ to $\theta_{\max}$ to calculate $\text{SLDR}(\theta_{\min})$ and $\text{SLDR}(\theta_{\max})$, as elaborated in Section 3.2. In Figure 13b, once again two intersections of $\widehat{\text{SLDR}}(\theta_{\min})$ and $\widehat{\text{SLDR}}(\theta_{\max})$ exist for this layer. Considering the constant distribution ($\frac{\partial \text{SLDR}}{\partial \theta} < lim_{\text{SLDR}}$) and high values of SLDR ($\text{SLDR}(\theta_{\min}) > \lim_{\text{pro}}$ and $\text{SLDR}(\theta_{\max}) > \lim_{\text{pro}}$), we can identify the intersection in the columnar particle shape class (see Table 2), resulting in $\xi > 1$ and $\kappa < -0.8$ as the most likely one. Figure 14 shows the vertical profile of $\xi$ which confirms the dominance of prolate particles in the investigated cloud. Accordingly, the Cloudnet classification, shown in Figure 11 (black box), classifies the hydrometeors before the RHI scan as supercooled liquid droplets, and after the RHI scan as ice-containing and partly mixed-phase layer down to about 1500 m height. A rain event occurs a few minutes after the RHI scan, defining drizzle or rain. The temperature of the investigated case ranges from $-3°$C at the cloud base and $-7°$C at the cloud top. This temperature range is characteristic for the formation of hydrometeors in the columnar particle shape class, which demonstrates the ability of VDPS to derive prolate particles.

### 4.3 Oblate particle shape class: Plate-like crystals on 4 January 2017 at 01:30 UTC

The third case study aims on the description of oblate particles, such as plate-like crystals. The corresponding measurement was recorded in Limassol on 4 January 2017 during an RHI scan from 01:31 to 01:33 UTC. The observed cloud system is marked by the black frame in Figure 15. The observation was characterized by the presence of a relatively homogeneous liquid-topped ice cloud in the height range from 3200 m to 4200 m. Figure 16 shows the RHI scan of SLDR from $90°$ to $150°$ elevation angle at 01:31 UTC. An increase of SLDR from $-30$ dB to $-10$ dB between $\theta_{\min}$ and $\theta_{\max}$ is visible. The linear regression is

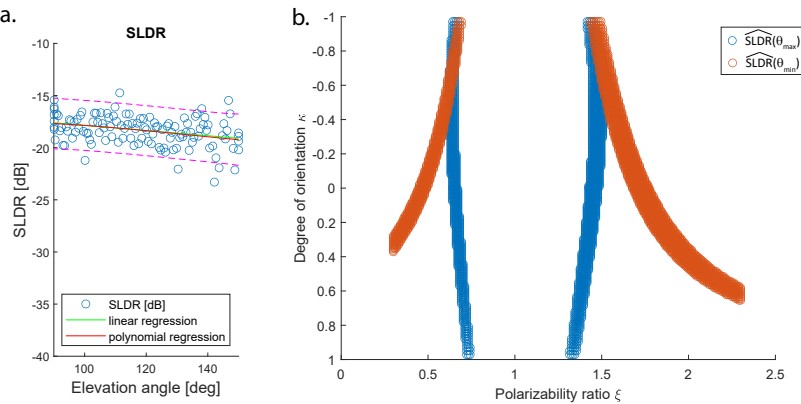

**Figure 13.** Detailed view into the columnar-shape case study presented in Figure 12 for the layer from 2458 m to 2490 m height. (a) Distribution of measured values of SLDR from $\theta_{min}$ to $\theta_{max}$ elevation angle and associated linear and polynomial fits. The dashed pink line in (a) corresponds to the 95% prediction interval from the third degrees polynomial function, used to determine the intersection of $\widehat{\mathrm{SLDR}}(\theta_{min})$ and $\widehat{\mathrm{SLDR}}(\theta_{max})$. (b) Intersection between $\widehat{\mathrm{SLDR}}(\theta_{min})$ and $\widehat{\mathrm{SLDR}}(\theta_{max})$ at $\theta_{min}$ and $\theta_{max}$, respectively.

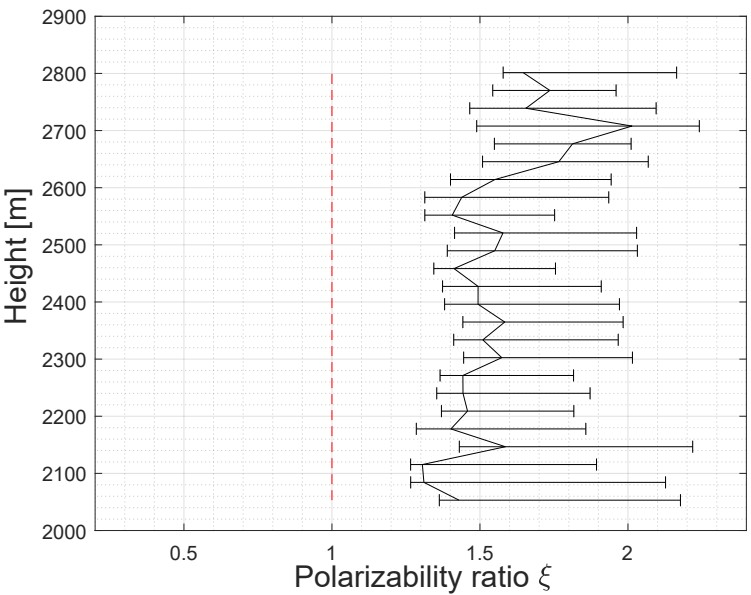

**Figure 14.** Polarizability ratio $\xi$ calculated for each layer with the VDPS-method for the columnar-shape case study, observed at Limassol on 8 December 2016, at 00:31 UTC.

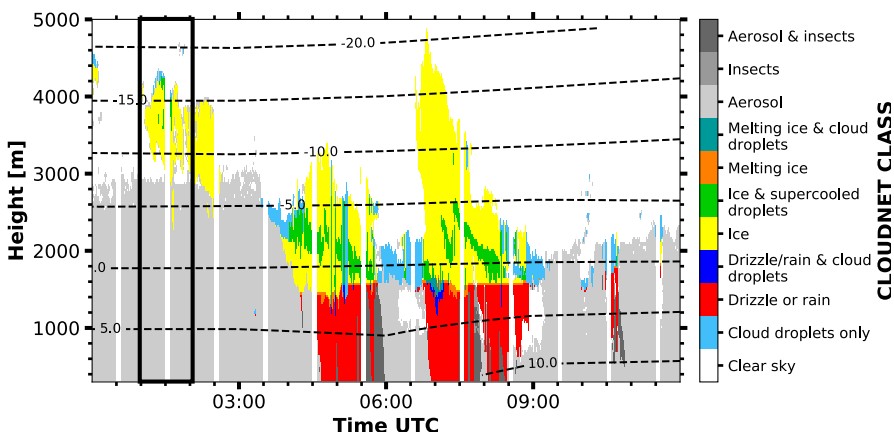

**Figure 15.** Cloudnet target classification mask as derived for observations at Limassol on 4 January 2017 from 00:00 to 12:00 UTC. The black box denotes the RHI scan that is discussed in further detail in Section 4.3.

represented by the green line in Figure 17a which exemplarily shows the retrieval for the layer from 3300 m to 3331 m height, represented by the black line on the y-axis in Figure 16. In this case, $\frac{\partial \mathrm{SLDR}}{\partial \theta} > lim_{\mathrm{SLDR}}$. $\mathrm{SLDR}(\theta_{\min})$ and $\mathrm{SLDR}(\theta_{\max})$ are calculated based on the values retrieved from the polynomial fit at $\theta_{\min}$ and $\theta_{\max}$, i.e., the red curve represented in Figure 17a. In Figure 17b, we see two intersections between the isolines of $\widehat{\mathrm{SLDR}}(\theta_{\min})$ and $\widehat{\mathrm{SLDR}}(\theta_{\max})$. This configuration, associated

430    with a positive linear regression of the polarimetric parameter SLDR (see table 2), implies to select the intersection at $\xi < 1$ and $\kappa > 0.8$ for determination of the exact polarizability ratio, which corresponds to the oblate primary particle shape class. The vertical distribution of $\xi$ presented in Figure 18 indicates $\xi < 1$ for all layers in the investigated cloud. The values of $\xi$ are relatively constant around 0.4 from 3100 m to 3600 m height corresponding to particles which are strongly oblate and rather dense, pointing likely to the class of thick plate crystals (Reinking et al., 2002; Matrosov et al., 2012). On the other hand, above

435    3600 m height, $\xi \approx 0.55$ was observed, representing particles which are likely less dense such as plates or dendritic crystals. In the Cloudnet classification shown in Figure 15, where the period of approximately 1 hour around the investigated RHI scan is indicated by the black rectangle, ice crystals and contributions of supercooled liquid droplets at cloud top were identified. The temperature in the cloud ranges from $-15°C$ at cloud top to $-10°C$ at cloud base. Laboratory studies suggest that, in this temperature range, the primary formation of plate-like ice crystals is most likely to occur (Bailey and Hallett, 2009). Hence,

440    there is a remarkably good agreement between results of the VDPS method and observations for this case study, as well.

### 4.4    Microphysical transformation: case study from 2 February 2017, 13:31 UTC

By means of a final case study, the potential of the VDPS method for exploration of the vertical evolution of particle shapes from cloud top to cloud base is discussed. The corresponding measurement was recorded in Limassol on 2 January 2017. In Figure 19, the Cloudnet target classification mask of the observed cloud system is shown. The black frame in Figure 19

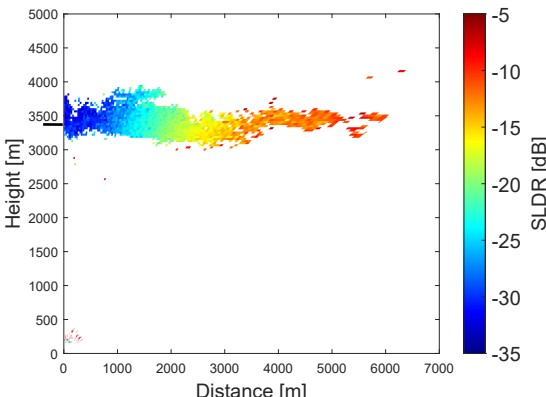

**Figure 16.** RHI-scan of SLDR on 4 January 2017, at 01:31 UTC in Limassol from $90°$ to $150°$ elevation angle. The black horizontal lines on the y axis mark the height of the layer analysed in Figure 17.

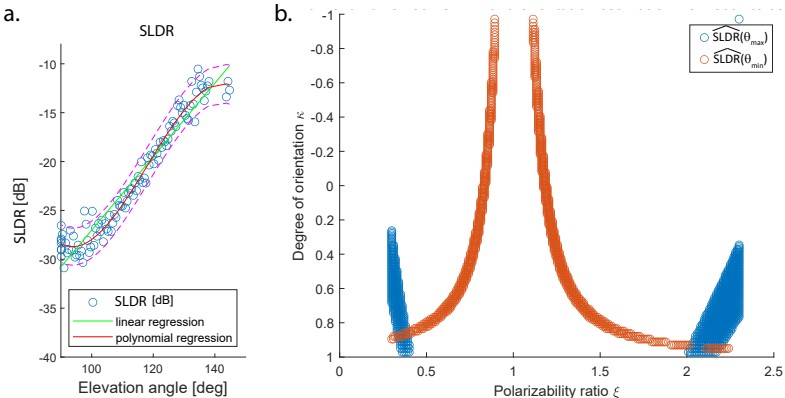

**Figure 17.** Detailed view into the plate-like-shape case study presented in Figure 8 for the layer from 3300 m to 3331 m height. (a) Distribution of measured values of SLDR from $\theta_{min}$ to $\theta_{max}$ elevation angle and associated linear and polynomial fits. The dashed pink line in (a) corresponds to the $95\%$ prediction interval from the third degrees polynomial function, used to determine the intersection of $\widehat{\text{SLDR}}(\theta_{min})$ and $\widehat{\text{SLDR}}(\theta_{max})$. (b) Intersection between $\widehat{\text{SLDR}}(\theta_{min})$ and $\widehat{\text{SLDR}}(\theta_{max})$ at $\theta_{min}$ and $\theta_{max}$, respectively.

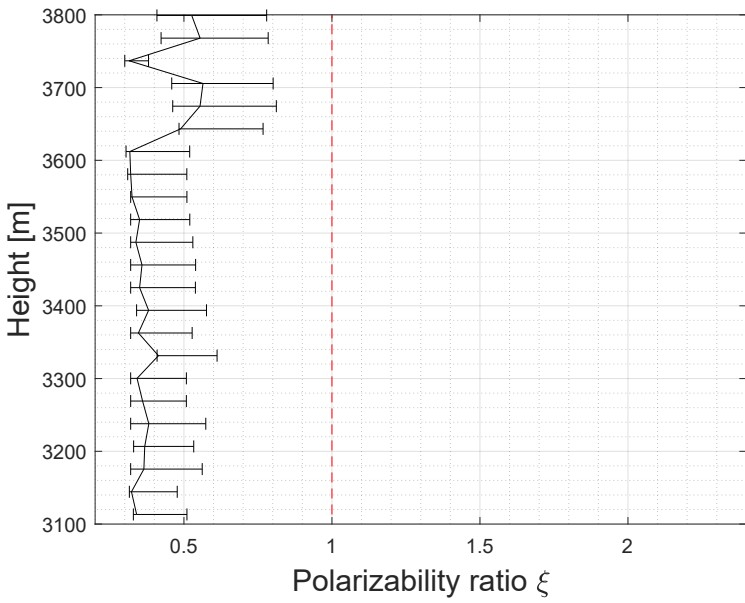

**Figure 18.** Polarizability ratio ξ calculated for each layer with the VDPS method for a plate-like-shape case study observed at Limassol, on 4 January 2017, at 01:31 UTC.

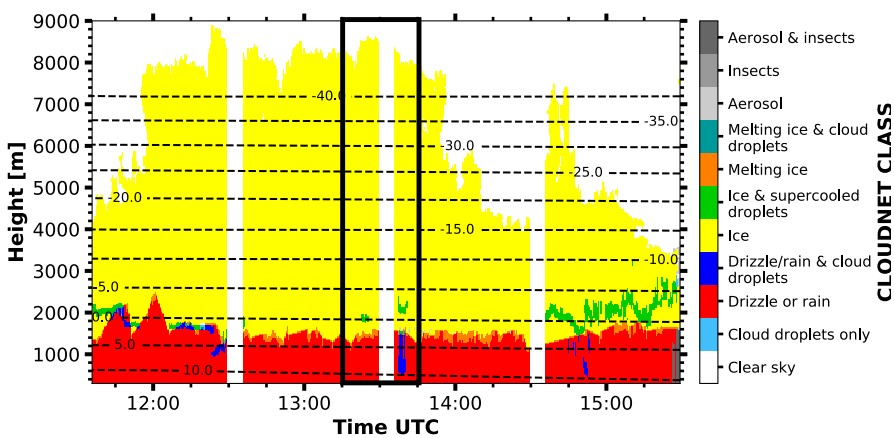

**Figure 19.** Cloudnet target classification mask as derived for observations at Limassol on 12 February 2017 from 11:30 to 15:30 UTC.

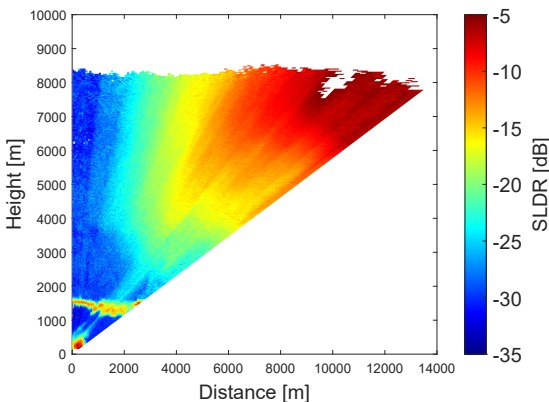

**Figure 20.** RHI scan of SLDR from $90°$ to $150°$ elevation angle observed in Limassol on 2 January 2017, 13:31 UTC.

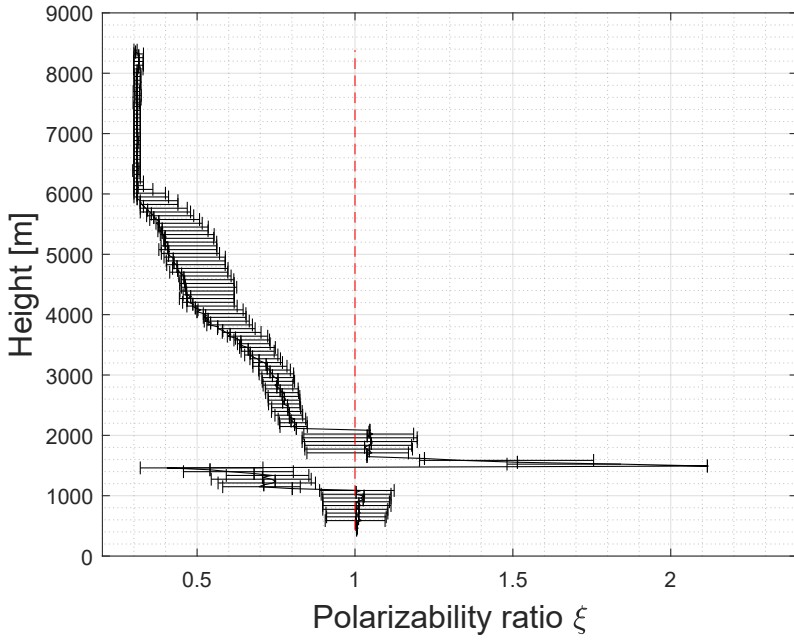

**Figure 21.** Profile of the polarizability ratio $\xi$ on 2 January 2017, 13:31 UTC, Limassol, as obtained from the RHI scan of SLDR presented in Fig. 20.

highlights the time period around the RHI scan at 13:31 (white vertical bar), which will be analysed below. As can be seen from the Cloudnet classification, ice crystals were identified at all heights from cloud top (around 8500 m) down to the melting layer, which was classified at a height of around 1700 m. Only at heights between around 2000–2500 m, few data points of mixed-phase conditions were identified. Also in Figure 20, which shows the 13:31 UTC RHI scan of SLDR from 90° to 150° elevation angle, the melting layer is well represented at around 1700 m height by increased values of SLDR at all elevation angles. Focusing on the height range above the melting layer, the elevation dependency of SLDR shows a distinct evolution from cloud top to bottom. At the top of the cloud, at around 8000 m height, we can observe a strong increase of SLDR from $\theta_{\min}$ to $\theta_{\max}$ ($-30$ dB at $\theta_{\min}$ and $-5$ dB at $\theta_{\max}$). Moving away from the cloud top towards the melting layer, the increase of SLDR from $\theta_{\min}$ to $\theta_{\max}$ becomes gradually less pronounced. Slightly above the melting layer ($\approx 2000$ m height), SLDR assumes values of around $-30$ dB at all elevation angles. The gradual change of the elevation dependency of SLDR from cloud top to cloud base translates into the vertical distribution of the polarizability ratio, as is illustrated in Figure 21. From 8000 m to 2000 m height, the polarizability ratio $\xi$ increases gradually from 0.3, corresponding to very oblate and dense particles, such as plates, to 0.8 corresponding to less dense oblate particles such as dendrites or aggregates. Between 2000 m height and the melting layer, located at 1700 m height, the polarizability ratio $\xi$ is close to 1 corresponding to particles with low density or generally spherical particles. This gradual increase in $\xi$ informs about a vertical change in particle shape while the ice crystals sedimented through the cloud system. As outlined earlier, a direct determination of the the types of microphysical processes that occurred in this case cannot be achieved, as further constraints must be incorporated for a thorough interpretation as is outlined in Section 5.

## 5  Discussion and conclusions

In this article, the Vertical-Distribution-of-Particle-Shape (VDPS) method was introduced. Based on earlier studies, which have succeeded in demonstrating the applicability of polarimetric parameters from cloud radar to estimate the particle shape (Matrosov et al., 2012; Myagkov et al., 2016a), this new approach aids one to characterize the shape of cloud particles from scanning SLDR-mode cloud radar observations. The new VDPS method is based only on a single polarimetric parameter - SLDR. Another novelty of the VDPS method is the idea that a profile of the polarizability ratio can be used not only to derive shape of pristine ice crystals at cloud tops (as done in Myagkov et al. (2016a, b)) but also as an indicator of microphysical processes affecting particle shape and/or apparent density in deep precipitating clouds. In addition, the VDPS method is more versatile than the original approach of Myagkov et al. (2016a), which was developed for hybrid-mode cloud radars, requiring a complex calibration of ZDR and correlation coefficient. We will compare the two methods in an upcoming campaign in Switzerland (winter 2023/24), where an SLDR (Metek S/N MBR5) and a hybrid-mode radar (Metek S/N MBR7) will operate co-located next to each other.

The $45°$-slanted linear depolarization (SLDR) mode was specifically chosen for the purpose of minimizing the influence of fluctuations in the particle orientation during sedimentation, called wobbling effect (Matrosov et al., 2001), while providing well suited and relatively easy observable input parameters for the shape retrieval. The VDPS approach represents a new,

versatile way to study microphysical processes by combining a spheroidal scattering model (Myagkov et al., 2016b) applied only to $\widehat{\mathrm{SLDR}}$. In this paper, the VDPS method was introduced and validated by means of case studies collected in the frame of the CyCARE field campaign (Limassol, Cyprus), for three representative shape classes oblate, isometric and prolate particles, which are characterized by polarizability ratios of $\xi < 1$, $\xi = 1$ and $\xi > 1$, respectively. A fourth case study demonstrated the potential of the VDPS method for tracking of the evolution of the ice crystal shape between top and base of a deep cloud system. Before application of the VDPS method to the case studies, the algorithm was tested and calibrated with success based on observational datasets from two field campaigns, CyCARE in Limassol, Cyprus, and DACAPO-PESO in Punta Arenas, Chile (Radenz et al., 2021), which sums up to three years of SLDR measurements at two different places. It is important to highlight that we could not validate the method using in-situ observations throughout the two campaigns. It is nevertheless the goal of the authors of this study to aim on deployments of the SLDR-mode scanning cloud radar in campaigns where in-situ observations are available.

The vertical distribution of the polarizability ratio $\xi$ is precious because it informs about the transformation of apparent particle shape or density in an investigated cloud from top to bottom, which shows that microphysical processes are occurring. Based on the information about the vertical distribution of particle shape in a cloud, the VDPS method provides valuable constraints for microphysical fingerprinting studies (Section 4.4). The height-resolved view of the vertical distribution and evolution of particle shape in a cloud is helpful to study and characterize mixed-phase cloud processes in the onset phase of precipitation. While isometric, columnar and oblate particle shapes can well be distinguished with the VDPS method, discrimination between graupel (formed by riming) and aggregates (formed by aggregation) remains a challenge and is currently not possible solely with the VDPS method. Nevertheless, both processes can potentially be inferred based on the vertical evolution of $\xi$ between cloud top and cloud base. In future, we therefore plan to associate the VDPS method with Doppler spectral methods in order to detect supercooled liquid droplets in mixed-phase clouds and to estimate the fall velocity of particles, which provide relevant constraints for the discrimination between riming and aggregation processes. Indeed, riming processes require the presence of supercooled liquid droplets and the formed graupel are falling faster than aggregates because of their higher density (Kneifel et al., 2016; Vogl et al., 2022).

Besides the mentioned strengths of the VDPS method, there are also certain limitations, which can eventually be overcome in future development steps. The first one is corresponding to the radar antenna quality, as it determines the calibration of SLDR. The polarimetric parameter SLDR is intrinsically dependent on the calibration of the antenna and the differential phase of the transceiver unit. Care must be taken to ensure a good calibration of the radar system. A good co-cross channel isolation should be aspired in order to obtain highest accuracy of the retrieval, especially for values of $\xi$ that are close to 1. In addition, turbulence, horizontal wind, and radar beam width, especially at large off-zenith pointing angles, can lead to a broadening of the Doppler spectra, which has the potential to impact the spectral peak values in both channels (Kollias et al., 2011). Spectral broadening becomes noteworthy when particles with distinct polarimetric signatures are blended into a single spectral line, and it becomes particularly relevant when substantial turbulence is present (typically on the order of several meters per second). However, the spectral broadening would not considerably change observed polarimetric signatures in the case of pristine ice crystals at the cloud top, or when only one type of hydrometeors is present in a cloud volume. Finally, in our

study we are assuming Rayleigh scattering and describe particle shapes according to the aspect ratio and the permittivity. In reality, ice crystal shapes are more complex and need a more sophisticated scattering method to accurately capture scattering of particles with axis lengths exceeding the range of the Rayleigh scattering regime. This holds definitely true for absolute quantities such as reflectivity at wavelengths shorter than C-band (Lu et al., 2016; von Terzi et al., 2022). However, a recent study of Matrosov (2021), demonstrates that the influence of non-Rayleigh scattering is weak for polarimetric variables such as LDR. As a likely reason for this behaviour, Matrosov (2021) hypothesizes that polarimetric variables are differential (rather than absolute) quantities representing differences/ratios of radar parameters at two orthogonal polarizations. T-Matrix or DDA methods provide much more degrees of freedom concerning the microphysics of the scattering hydrometeors. If these are applied to realistic hydrometeor populations, a model-based validation of the hypothesis of Matrosov (2021) shall be feasible.

Secondly, in its current development state, the VDPS method is also only capable to investigate the shape of the hydrometeor population that determines the main peak of the co-channel Doppler spectrum, as characterized by the highest peak of each Doppler spectrum obtained during an RHI scan at any given height level. However, a new approach taking into account the comparison between main peaks detected in the co- and cross-channels can give more information about the ice crystal populations in a volume: if the main peaks are similar in the co- and cross-channels, it means that the main hydrometeor population depolarizes the most. On the other hand, the presence of different main peaks in the co- and cross-polarized Doppler spectra would imply the presence of a second hydrometeor population which depolarizes strongly, while still a non-polarizing hydrometeor population dominates the co-channel signal.

The technique can currently thus not be used for evaluating the RHI scans for coexistence of several particle populations, as they might be superimposed by means of their differential fall velocities collected in a Doppler spectrum. Such peak separation techniques have already been developed for vertically pointing cloud radar measurements (Kalesse et al., 2019; Radenz et al., 2019) and can potentially be adapted for scanning cloud radars in the near future.

Overall, the VDPS technique has the potential to become a standard procedure in the analysis of long-term observations from scanning SLDR cloud radar systems. Given the broad availability of scanning LDR-mode cloud radars in Europe, the VDPS method provides good reasoning to update these to SLDR mode with low effort and investment.

*Code and data availability.* The cloud-radar raw data and retrieval codes are available upon request. Please contact the first or second author. Cloudnet data are available at https://cloudnet.fmi.fi. For plotting of the data, the tool pyLARDA, available at https://github.com/lacros-tropos/larda, was used.

*Author contributions.* AT developed the VDPS method, analysed the data and drafted the manuscript supervised by PS. JB conducted the CyCARE campaign and operated LACROS. PS, MR, and JB generated the Cloudnet datasets and supervised the data processing chain. AM suppored me during the starting phase of the development work on the VDPS method and developed the spheroidal scattering model.

*Competing interests.* The contact author has declared that neither they nor their co-authors have any competing interests.

*Acknowledgements.* Development of the VDPS method was funded by the Deutsche Forschungsgemeinschaft (DFG − German Research Foundation) project PICNICC (SE2464/1-1 and KA4162/2-1). The authors wish to thank Cyprus University of Technology, Limassol, Cyprus, for their logistic and infrastructural support during the LACROS deployment. We gratefully acknowledge the ACTRIS Cloud Remote Sensing Unit for making the Cloudnet datasets publicly available. LACROS operations were supported by the European Union (EU) Horizon 2020 (ACTRIS; grant no. 654109) and the Seventh Framework Programme (BACCHUS; grant no. 603445). The authors also wish to thank Metek GmbH, Elmshorn, for the technical support related to the Mira-35 radar.

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
