# Peer review of "Determination of the vertical distribution of in-cloud particle shape using SLDR mode 35-GHz scanning cloud radar"

_EGUsphere, 2022_

## Author Comment (AC2)

**Response to the Editor**

Dear Editor,

First of all, we want to thank you for your and the reviewers' valuable suggestions, which have greatly improved or article. Then, we want to apologize for the difficulty encountered by the reviewers in understanding the first version of the manuscript. We have therefore reworked the manuscript, in particular section 3, so that it becomes more accessible for the reader. We have five main changes in the revised manuscript: (1) We decided to calculate SLDR from the maximum spectral lines in the co-channel (and not in the cross-channel) and (2) to simplify the method using SLDR only and removing $\rho_{cx}$ in order to make it more accessible to people which are less familiar with polarimetric parameters. (3) We changed the notation of $\delta_s$ to SLDR to be more consistent with the one used in earlier studies, e.g., Myagkov et al., 2016. (4) We selected a new columnar case study in Section 4.2 which is focusing on a thinner cloud layer and (5) we added a new case study in Section 4.4, showing the ability of the VDPS method to detect microphysical processes by a vertical particle shape change. In addition to our responses to the reviewer comments, we applied a few further modifications to the manuscript, which address mostly the figure layout and the improvement of readability. All of them are also highlighted in the diff version of the resubmission. We hope that the overall sum of the applied modifications will lead to an improved understandability of the article and, therefore, make it suitable for publication in your journal.

Consequently, we would like to submit the revised manuscript and the diff-version of the revised manuscript together with our responses to all the comments provided by the two reviewers. In our replies, all references to modified lines are given with respect to the final revised version of the manuscript (without differences). Thank you for considering our work,

Best regards,

Audrey Teisseire, Patric Seifert, Alexander Myagkov, Martin Radenz and Johannes Bühl.
teisseire@tropos.de

**Response to reviewer 1# :**

**Comment 1 :**

Therefore, I believe the authors should carefully discuss the research progress of this field and highlight the contribution of this work, instead of simply referring to for example Myagkov et al. (2016a).

**Response 1 :**

The VDPS method is a new versatile version of the method developed in Myagkov et al., 2016a. As explained in the introduction of this letter, we decided to simplify the method removing the cross-correlation coefficient from the method. Indeed, the calibration of $\rho_{cx}$ is strongly dependent on the antenna quality which is not compatible with the spheroidal scattering model developed in Myagkov et al., 2016a, and used in the VDPS method (see Section 3.2). In the revised manuscript we propose a new version of VDPS method, working with SLDR only.

Our main motivation for the study is defined by the availability of scanning LDR-mode cloud radars in Europe, which potentially with a very little effort and investment can be upgraded to the SLDR-mode. Such an upgrade would allow for gathering a long-term dataset with a possibility of an advanced characterization of ice particles. The two methods detailed in Myagkov et al., 2016, and Teisseire et al., 2023, will be compared in a upcoming field campaign in Switzerland where an SLDR (Metek MBR5) and a hybrid-mode (Metek MBR7) radar will be co-located.

We added in the conclusion Section, lines 496-498 : "*Overall, the VDPS technique has the potential to become a standard procedure in the analysis of long-term observations from scanning SLDR cloud radar systems. Given the broad availability of scanning LDR-mode cloud radars in Europe, the VDPS method provides good reasoning to update these to SLDR mode with low effort and investment*"

.**Comment 2 :**

Different observing modes were mentioned in the manuscript, but no explanations were presented. It would be beneficial to summarize the characteristics of different modes in a table, and show why and how this mode is superior to others.

**Response 2 :**

Yes, it is only mentioned in the manuscript that SLDR-mode (which is LDR-mode but slanted at 45°) is more sensitive to particle shapes. Indeed, due to the rotation of the antenna around the emission source, the received polarization will be less impacted by the wobbling effect of particles. In this manner, for our study, it is more relevant to use the polarimetric parameter SLDR to describe particle shapes. Some studies explained well the difference between radars (see the poster from the Metek company below , https://www.google.com/url?sa=t&rct=j&q=&esrc=s&source=web&cd=&cad=rja&uact=8&ved=2ahUKEwiz0N6W5ur_AhXizgIHHchVBOcQFnoECBYQAQ&url=https%3A%2F%2Fams.confex.com%2Fams%2F38RADAR%2Fwebprogram%2FHandout%2FPaper321249%2FPosterPolarization.pdf&usg=AOvVaw1KU9q4bPMgHEm6hykFyf-_&opi=89978449).
First, we attempted to provide the requested table, but we quickly realized that the introduction and discussion of different radar modes would go beyond the scope of our study. We thus decided to provide a reference to the book of Bringi, 2009, Chapter 6, to line 89 : "*Numerous polarimetric configurations of radar systems exist (Bringi, 2009, Ch. 6), but the LDR mode is currently the most common one amongst cloud radars.*"

**Comment 3 :**

I am frustrated in connecting the symbols used this work to existing works. For example, SLDR is denoted by delta_s, rho_hv by rho_s. And polarization ratio, orientation…I took very long time to connect them to what have been used by Myagkov 2016. Other readers may have similar feelings… I would suggest keeping consistent with Myagkov who is the coauthor of this work.

**Response 3 :**

We find the proposition reasonable to use SLDR instead of $\delta_s$ to be coherent with the work of Myagkov et al., (2016) and Matrosov et al., (2012). Some co-authors work in lidar research,

where single greek letters are used in mathematical expressions, which motivated us to introduce the single-letter parameters. Concerning $\rho_s$, we decided to remove this parameter from the method, as explained in response 1.

**Comment 4 :**

In addition, I feel the first paragraph in section 2.3 is poorly organized. Please elaborate the used variables before analyzing the results in Fig.2.

**Response 4 :**

The reviewer is right, the Section 2.3 was rephrased and reorganized by the authors in the revised manuscript to make it more understandable.

**Comment 5 :**

I believe that the Rayleigh condition at Ka-band should be considered. The theoretical basis is based on the assumption of Ka-band Rayleigh scattering which is satisfied for pristine ice. However, in presence of large aggregates which usually occur at -5 to 0 C or -15 C, this assumption is violated and the retrieval should be made with caution.

**Response 5 :**

The question was also asked by the second reviewer, please see the details in the response I.3, below in the second part.

**Comment 6 :**

L160-163. the concept of orientation should be well discussed. The reasons why these assumed K values are applicable should be elaborated.

**Response 6 :**

The reviewer is right, it was not really clear in the section 2.3. The degree of orientation of particles is not assumed but retrieved by the spheroidal scattering model. The degree of orientation is well explained in Myagkov et al., 2016a by the formula 11 and the figure 9 of this paper. We added this reference to the revised manuscript, see lines 173-174 : "*The degree of orientation characterizes the width of the particle orientation angle distribution (the degree of orientation is explained in more details in Myagkov et al., (2016), in Figure 9 and Equation 11 in there)*". In our paper, VDPS method doesn't retrieve the degree of orientation and we are focusing on the particle shape described by the polarizability ratio. That is why the VDPS method provides as output the vertical distribution of the polarizability ratio, only. In this study, the degree of orientation is used quantitatively only.

We deleted the following sentence from Section 2.3 of the original submission: "*In contrast, as a comparison of Figures 2a and 2b shows, ρs is a proxy of degree of orientation κ (as described by Ryzhkov (2001)). Indeed, ρs features no sensitivity with respect to ξ, as gradients of ρs against the elevation angle only exist with respect to κ.*".

**Comment 7 :**

The authors claim that rimed or aggregated ice particles are isometric and speculate that the retrieved polarization ratio around 1 indicates riming or aggregation. I feel that this statement should be used with caution. Firstly, the method may be limited to pristine ice at ka band. The Rayleigh condition at Ka band may not have reached, for example at 1.6 km in figure 10. Therefore, the basic assumption is broken. Secondly, very heavily rimed ice can be isometric. Lightly rimed ice or aggregation has a characteristic aspect ratio.

**Response 7 :**

Apologies for the misunderstanding. The polarizability ratio is depending on the axis ratio (which is the indicator of particle shape) and the ice density. It is known that ice density is typically lower in aggregates, therefore, polarimetric signatures are less pronounced for aggregates and rimed particles. For this reason, the polarizability ratio will be close to 1 for aggregates, because of the combination of low density and close-to-unity axis ratio.

From the polarimetric observations we cannot directly classify processes like riming and aggregation and we cannot say at which altitude these processes happen, but from the polarizability ratio profile we can see that something happens, a process changing shape and/or density of particles (see an example in Section 4.4, in the revised manuscript). This alone is a valuable information which a conventional vertically pointed and single frequency radar cannot give.

We changed the content of lines 371 : "*as both of which form isometric graupel particles or aggregates, respectively*" to "*which cannot unambiguously be identified solely with the VDPS method.*" And we added lines 362, 368 and 370 : "*... isometric or less dense particles*".

**Comment 8 :**

I understand that the authors do not have aircraft observations at hand for validation. Why the retrieved results are consistent with cloud physics should be discussed and the weaknees that no direction comparison was done should be discussed.

**Response 8 :**

We agree to the reviewer, that a direct evaluation of the retrieval technique should be preferred and that the current evaluation approach is actually a consistency check. Unfortunately, we do not have in-situ data (and in general it is hard and expensive to have aircraft or general airborne measurements), therefore, the presented method is based on a STSR-mode retrieval which has been verified by a joint analysis of radar observations of ice particles formed at top of mixed-phase clouds and ice particles grown under similar conditions in the lab (Myagkov et al.,2016a). Secondly, the temperature is precious to describe ice particle formation. If you have a look at the Figure 1 from the reply letter, we can see that between -3°C and -10°C, only columns can be formed while between -10°C and -20°C only plate-like are susceptible to form in this condition. To validate the VDPS method, we chose case studies according to this assumption (see Section 4.2 and 4.3), which represents a relevant evidence. The second case study representing columnar crystals has been changed one featuring a thinner cloud layer (see Section 4.2).

[Figure]

*Figure 1 : Classification of ice particles depending on the temperature and the supersaturation (Kenneth G. Libbrecht, 2012).*

**II. Technical issues**

**Comment 9 :**

L109. Do you mean that the spectral line with the highest Scx was used? If so, a figure illustrating the spectral processing is needed, since this has rarely been done.

**Response 9 :**

Yes, we used the maximum spectral lines in the cross-channel to derive SLDR. In general, quality of polarimetric observations depends on SNR (Bringi and Chandrasekar, 2001). In LDR and SLDR radars SNR in the co-channel by several orders of magnitude exceeds SNR in the cross-channel. Thanks to the reviewer comment, we decided to recalculate SLDR with the maximum spectral lines in the co-channel (See Section 2.1). By doing this, we improved drastically the quality of Figures in the revised manuscript and we realized the potential of comparing SLDR calculating in the co- and cross-channel. We added in the conclusion Section, lines 486-491 : "*However, a new approach taking into account the comparison between main peaks detected in the co- and cross-channels can give more information about the ice crystal populations in a volume: if the main peaks are similar in the co- and cross channels, it means that the main hydrometeor population depolarizes the most. On the other hand, the presence of different main peaks in the co- and cross-polarized Doppler spectra would imply the presence of a second hydrometeor population which depolarizes strongly, while still a non-polarizing hydrometeor population dominates the co-channel signal.*"

**Comment 10 :**

In addition, have you quantified the impact of spectral broadening which can effectively smooth the spectral peak?

**Response 10 :**

Spectral broadening has no impact on the SLDR calculation with this method calculating SLDR from the maximum spectral line in the co-channel (see response II.9). This effect could smooth the spectral peak, but symmetrically, that means that the highest value of Sco remains unchanged.

**Comment 11 :**

Fig. 3. add discussions about orientation.

**Response 11 :**

Figure 3 is focusing on the distribution of SLDR from 90° to 150° elevation angle. The values of SLDR are processed for the three particle shape classes (oblate, isometric and prolate particle shape classes) represented by a specific symbol in Figure 2. That is why we don't need to discuss about the degree of orientation in Figure 3. We hope that this notion will be easier to understand by removing $\rho_{cx}$ from the method.

**Comment 12 :**

Fig. 6. Orientation information is missing

**Response 12 :**

Yes, you are right, we added in the caption of Figure 6 : "*SLDR(θmin) = −32dB, SLDR(θmax) = −11dB, and κ = 0.85*".

**Comment 13 :**

L174-175. I do not see the discussions about dendritic in your figures.

**Response 13 :**

Dendrites are considered as oblate particles in the paper. We changed dendrite to plate-like in the revised manuscript, to be more general. We added line 193 : "*Finally, plate-like particles, belonging to the oblate particle class,…*" to specify the oblate aspect of dendrites.

**Comment 14 :**

L178-180. The discussion on isometric particle shape class is lack of ground.

**Response 14 :**

We rephrase and reorganize the Section 2.2 to be more understandable. We added lines 192-193 : "*The isometric primary particle shape class is represented by constantly low values of SLDR at all elevation angles between 90° and 150°.*".

**Response to reviewer 2# :**

**I. First observations**

**Comment 1 :**

The authors should make more clear which parts of this work are substantially different from the previous approach by Myagkov et al., 2016a,b. I understand that the hybrid-mode Ka band radar in Myagkov et al., 2016 is different from the SLDR mode radar in this study. But as mentioned in Myagkov et al., 2016a their approach can also be applied to SLDR mode radars. So what are the major changes and improvements in this work?

**Response 1 :**

Our main motivation for the study is defined by the availability of scanning LDR-mode cloud radars in Europe, which potentially with a very little effort and investment can be upgraded to the SLDR mode. Such an upgrade would allow for gathering a long-term dataset with a possibility of an advanced characterization of ice particles. The reviewer is right that the original method of Myagkov et al., 2016 can be applied to SLDR mode radars, but it requires an accurate polarimetric calibration. This calibration is not so straightforward for the correlation coefficient $\rho_{cx}$ and we think that this may limit the acceptance of the method in a broad community especially for people who are not experts in cloud radar polarimetry. Therefore, we decided to develop a much simpler method, based, in the current state, only on a single polarimetric parameter - SLDR. Another novelty presented in the manuscript is an idea that a profile of the polarizability ratio can be used not only to derive shape of pristine ice crystals at cloud tops (as done in Myagkov 2016a,b) but also as an indicator of microphysical processes affecting particles' shape and/or apparent density in deep precipitating clouds. This information has been added into the introduction Section. With the revision we decided to readapt the retrieval and to make it applicable using SLDR only, which simplifies the understanding of the method for the reader while maintaining its applicability.

We modified the flowchart presented in Figure 4, added in the last step a new way to discriminate columns from isometric particles, where SLDR>-25dB for columnar crystals (see Section 3.3).

We added the following content to lines 64-68 of the introduction : "*Even though the number of scanning STSR-mode cloud radars has been continuously growing in Europe, a number of measurement sites within ACTRIS (the Aerosol, Clouds and Trace Gases Research Infrastructure) are equipped with scanning LDR radars (Madonna et al., 2013; Löhnert et al., 2015; Tetoni et al., 2022). Such radars can be modified to the SLDR mode with relatively low efforts and investments, and as a result provide long a term observational database for retrieving the polarizability ratio of ice-containing clouds in different climatic zones.*"

**Comment 2 :**

It would be also a great help for the reader if you could use more "common" notation for variables or make them at least consistent with the one used in Myagkov et al. Ideally, the method from Myagkov et al., 2016 should be directly compared to your new method using an identical case study, but I don't know whether such a dataset exists. I also think the authors need to describe better the state-of-the-art of different SLDR approaches and also provide all necessary references to previous work (SLDR work done by Reinking, Melnikov, Matrosov, etc.) to the reader (see comment to the introduction below).

**Response 2 :**

The reviewer argument convinced us, that we should not enhance the complexity of the technique by introducing a new notation. Some co-authors work in lidar research, where single greek letters are used in mathematical expressions. Concordantly, we changed $\delta_s$ to SLDR to be more coherent with the subject-related literature (Myagkov et al., 2016, Matrosov et al., 2012 …), and $\rho_s$ is removed from the method. This notation will facilitate the reading of this paper for people less experienced in the polarimetric radar field. Concerning the request for a comparative case study: unfortunately, we don't have an identical case study because we work with different campaigns and different radar types. Myagkov et al.,2016 uses the ACCEPT campaign with a STSR mode radar, while Teisseire et al., 2023 is written based on data from the CyCARE campaign, where a SLDR-mode scanning cloud radar was operated, see Section 2.1. We will compare the two methods in an upcoming campaign in Switzerland (winter 2023/24), when an SLDR (Metek MBR5) and a hybrid-mode radar (Metek MBR7) will operate co-located in order to obtain the evaluation dataset. We added in the discussion Section, lines 452-454 : "*We will compare the two methods in an upcoming campaign in Switzerland (winter 2023/24), where an SLDR (Metek S/N MBR5) and a hybrid-mode radar (Metek S/N MBR7) will operate co-located next to each other.*"

The comment related to the referencing of previous work is addressed further below in a dedicated statement.

**Comment 3 :**

The method described here and also the one in Myagkov et al., 2016a,b are based on Rayleigh scattering approximation. To my knowledge, it has not yet been proven (for example with DDA scattering simulations) that this approximation is a valid assumption for the polarimetric quantities used in this work or what the uncertainty range associated with the Rayleigh assumption is. Previous studies such as Schrom and Kumjian, JAMC, 2018 found quite large deviations in polarimetric variables for plate-like particles even when using T-Matrix and even at low radar frequencies. The fact that the retrieved values are consistent with laboratory data (as shown in Myagkov et al., 2016b) is in my opinion not sufficient as the agreement could be caused by compensating errors in the retrieval itself. I would have expected that the authors do at least an attempt to test the general consistency of the Rayleigh assumption with existing DDA scattering datasets (for example Lu et al., AMT, 2016). I also realized that the PICNICC project is part of a bigger German research collaboration on radar polarimetry (PROM). Wouldn't it be possible that some of the collaborating projects provide you with DDA calculations of plates and columns in order to check the validity of the Rayleigh assumptions which are the core of the method? This would have been a clear improvement or at least a confirmation of the previous approach.

**Response 3 :**

We thank the reviewer for this valuable comment. The algorithm indeed assumes Rayleigh scattering. It is true that large individual particles may produce scattering signatures different from those predicted by the Rayleigh spheroidal model. In reality, however, we never observe individual particles. There is a huge number of particles coexisting in the scattering volume. In the case of large aggregates or rimed particles, one of course expects resonance effects from individual particles and this is what DDA predicts. But in radar observations we almost never see these signatures. This is because no ice particles are exactly the same, they have different orientations and complexity which average out resonance effects in backscattering radar observables. And this (scattering from a mixture of many different ice particles) is what DDA databases do not currently have.

In rain where all droplets follow similar size-shape-velocity relations, polarimetric oscillations are nicely visible (Myagkov et al., 2020). However, in ice regions of deep clouds we rarely see polarimetric signatures produced by large ice particles. In order to support this point of view, we randomly took 80 deep precipitating clouds from the analyzed campaign, presented below, in Figure 2. For each cloud we took profiles of the attenuated reflectivity and SLDR, at 60° off-zenith angle. All profiles were then displayed on a single scatter plot given below. As one can see, even for the highest reflectivity values SLDR does not exceed -10 dB which roughly corresponds to ZDR of 0.9 dB. Of course, extreme cases such as in references given by the reviewer can still occur, but at least during the period of the Cyprus deployment of the SLDR cloud radar used in our study such cases are exceptional. Even in the case reported by Oue et al., 2018 JAMC, when ZDR of 7 dB was observed at X-band, the authors note that a simple spheroidal would indicate a presence of highly oblate particles. We would like to emphasize that the parameter we derive, polarizability ratio, is a function of the geometric axis ratio and apparent ice density. Please note, that in this study we propose to use this parameter as a proxy of changes in particle's microphysics but of course it is not possible to conclude which exact physical parameter (shape, density etc) changes. Luckily this decoupling is often not required. For example, if we have dendrites at the cloud top, the polarizability ratio will be well below 1. If throughout the cloud dendrites aggregate, this leads to aspect ratios closer to 1 and smaller apparent ice density. Both effects let the polarizability ratio approach values closer to 1. In the case of riming, particle's shape is getting closer to spherical, but density is increasing. However, the effect of density on the polarizability ratio is small in the case of shapes close to spheres and the polarizability ratio will become close to 1. Therefore, we think that the polarizability ratio can be used to detect such processes, although it is true that it is not possible to say which exact mechanism is responsible for changes of the polarizability ratio.

We added in Section 2.3, lines 154-163, this explanation : "*It is well known that the Rayleigh approximation is not always applicable to simulate scattering from large ice particles. Often the direct dipole approximation (DDA) is used to simulate scattering of individual ice particles having a complex shape. However, these simulations are often limited to a number of predefined shapes and therefore do not necessarily represent the ice particles observed by a radar. Simulations for a single particle also do not reflect the volumetric scattering effects. In general, ice particles in a scattering volume have arbitrary shapes and the contribution of individual particles to the backscattering radar observables is averaged out. We decided to assume the Rayleigh scattering and the spheroidal particle approximation (Matrosov, 1991; Ryzhkov, 2001; Bringi and Chandrasekar,2001) because (1) such a model explains general polarimetric scattering effects with just a few parameters, (2) the model parameters are well*

*constrained by the observations, (3) the volumetric scattering is taken into account, and (4) the model allows a computationally effective derivation of the polarizability ratio."*

[Figure]

Figure 2: Attenuated radar reflectivity Z and SLDR of 80 deep precipitating clouds as observed at 60° off-zenith angle in the course the CyCARE campaign.

**Comment 4 :**

As you mention in the text, one of the strong assumptions of all RHI based SLDR retrievals is that particle populations are homogenous at a certain height level within a few km horizontal distance. I was hoping that you provide some objective criteria for estimating the homogeneity of particles. Your method seems to still depend on the "experienced-eye" selection of RHI scans (L. 191). In my opinion, this is completely impractical for compiling statistics (which you mention in the conclusions L. 415 ff).

**Response 4 :**

In the algorithm we have a rule for homogeneity of clouds, namely at least 15% of data points in the RHI should be available (this limit number can be change depending on the radar calibration or scan velocity, as explained in Section 3). We assume that at a certain altitude conditions are similar and therefore the formation and evolution of ice particles is also similar. Please note that our study is based not on weather radars observing 100s km distance. The maximum distance of the used cloud radars is typically limited by 10 km. Thus, we assume horizontal homogeneity over a distance maximum of 8.5 km which is comparable with a footprint of spaceborne radars. Stratiform clouds typically have much larger scale and therefore we think that this assumption is correct in a majority of cases.

We added some sentences to Section 3, lines 209-214 : *"The scale of the horizontal homogeneity if defined by the maximum observation distance of the used cloud radar and the lowest elevation angle (10-15 km and 30°, respectively). Thus, the required scale of the horizontal homogeneity is mostly below 13 km, which is comparable e.g. to a footprint of spaceborne remote-sensing meteorological instruments. A majority of stratiform clouds have much larger spatial scale. In addition, the algorithm requires a minimum number of data points in each layer, representing 15% of data points, as will be explained in Section 3.1"*.

In our study, we decided to check RHI scans used for the four presented case studies with experienced-eyes (Section 4), and verify the homogeneity of the investigated clouds to validate the VDPS method. Statistics can be compiled with RHI scans previously chosen with experienced-eyes, but we decided to remove the following sentences about statistics from the Conclusion Section of the first submission: *"The VDPS method was implemented by means of an automatized framework, which permits us to obtain statistics about the particle shape for a long period of measurements and covering several field campaigns. In order to ensure the quality and the relevance of statistics, it is preferable to check the homogeneity of RHI scans beforehand with experienced-eyes."*

**Comment 5 :**

The biggest weakness in my view is the lack of an independent evaluation of the method. The case study analysis which you present is at the most a consistency check. The fact that the retrieved shape is roughly matching the expected shape in this temperature region cannot be counted as an evaluation of the method. How can you be sure that only one particle type with a distinct shape is present in your volume (the cloud in your second case is roughly 1.5km thick)? How can you know that your crystals are not partially rimed in an environment which Cloudnet classifies as mixed-phase? I think the case studies shown here are also less convincing than the comparison shown in Myagkov et al., 2016b where the authors selected only thin and clearly liquid-topped mixed-phase clouds.

**Response 5 :**

We agree to the reviewer, that a direct evaluation of the retrieval technique should be preferred and that the current evaluation approach in actually a consistency check. Please note, that a direct evaluation is nearly impossible to do for our study. Unfortunately, we do not have in-situ data (and in general it is hard and expensive to have aircraft measurements), therefore, the retrieval relies on the indirect evaluation given in Myagkov et al., 2016. The VDPS method does not aim to classify processes, and can only be used to detect the presence of a process changing shape and/or density without knowing the exact mechanism (Illustrated in Section 4.4).

Regarding a mixture of particles, we use the strongest spectral line in each co-channel (SNR) Doppler spectrum. Assuming that it is defined by the predominant hydrometeor type. Columns and plate-like particles have completely different size-velocity relations and therefore it is not likely that they have comparable reflectivity at the same velocity. In addition, distinct shapes (plates/columns) are formed at temperatures warmer than -20°C. For the temperature range considered in this study, it is known that only one major shape is formed by ice nucleation, as ice nucleation is only taking place via the liquid phase (Westbrook et al., 2011).

The reviewer is right about the second case study interpreted in Section 4.2, the cloud is too thick to be able to presume that only pristine crystals such as columns coexist. We decided to replace the columnar case study in Section 4.2 by one of a thinner cloud layer which is easier to interpret. Doing this, we have three case studies where the VDPS method can be evaluated for three particle shapes using the temperature range, responsible for the hydrometeor formation.

**II. Other general comments:**

I have two major criticism regarding the introduction:

**Comment 1 :**

a) In my understanding, an introduction should include an overview of the previous work done on the topic and also describe the current state-of-art. I miss a number of studies related to SLDR which can be easily found by a quick internet search. Currently, the introduction is heavily weighted toward the work of the authors and their working group. I strongly suggest that the authors provide a more thorough and less self-biased literature overview. For example in the paragraph starting at L. 44 the only two studies related to SLDR that you introduce are from Matrosov et al., 2012 and Myagkov et al., 2016a. This is not really complete considering the previous work done on this topic. No need to discuss every study but at least a complete list of references should be provided. Just a few examples of missing studies related to SLDR:

Reinking et al., JTECH, 2002 (Used SLDR to distinguish drizzle, crystals and irregular ice particles)

Matrosov et al., JAS, 2005 (Fall attitudes of dendrites derived using SLDR and HLDR)

Line 48 : we added the three references (Matrosov et al., 2001, Reinking et al., 2002 and Matrosov et al., 2005)

**Comment 2 :**

L. 41: The recent study by Luke et al., PNAS, 2021 where the authors used Doppler spectra and LDR to separate different hydrometeor populations is missing.

**Response 2 :**

We added the reference Luke et al., 2021 to line 42.

**Comment 3 :**

b) My second criticism is related to the motivation of the relevancy to retrieve shape information of ice particles. Just to avoid misunderstandings: I agree that information about ice particle shape is valuable but I find your discussion and argumentation of why this information is relevant for models or our microphysical understanding to be quite vague. For example, in L. 29-31: You argue that the shape information can be used to distinguish pristine ice from aggregates or rimed particles. Many aircraft studies showed in the past that most ice and mixed-phase clouds are dominated by irregular particles. (1) Can your method distinguish between irregular particles and aggregates/rimed particles? (2) What would

happen if you probed a mixture of columnar and plate-like particles? They could be both "pristine" but probably would look like aggregates. As you write in the introduction, if you are looking at the top of a mixed-phase cloud, you can assume that all particles grow into a particular shape due to the similar temperature regime they were nucleated. (3) But for deeper clouds, I highly question whether an approach indicating a change in shape can really distinguish between a mixture of ice crystal shapes, aggregation or riming. In L. 37-40 you also mention that the shape information could be useful to "support the improvement of these processes in numerical models". (4) Can you explain better how you think a vertical profile of shape can be useful for model development? I am very skeptical since most bulk schemes assume a constant ice particle habit throughout the entire temperature range. (5) Only very few experimental models currently exist which take particle habit into account.

**Response 3 :**

Thank you for these interesting questions. We will answer them one by one to be as efficient as possible.

(1) The VDPS method is able to determine the particle shape of the main population of hydrometeors for each layer. Concerning aggregates, they will have a lower density compared to pristine crystals (with a polarizability ratio near to one), while graupel particles have a higher density and a rather spherical shape (with a polarizability ratio near to one). The VDPS method is supposed to probe the overall vertical distribution of particle shapes - as the VDPS acronym already indicates – for the predominant hydrometeor population. We want to observe the starting phase and the transition from a certain initial particle shape to the shape parameter observed at another point in the cloud. This approach will help in the utilization and interpretation of further retrieval techniques. E.g., the ice crystal number concentration retrieval of Bühl et al., (2019, AMT), requires that the particle shape is known. Appropriate retrievals are only possible where the crystal population can be assumed to be homogeneous. VDPS is planned to provide anchor points for the application of the Bühl-2019 ICNC retrieval. Ice particles evolve due to sedimentation, deposition, sublimation, aggregation, and riming. A vertical change in shape or density can be due to a microphysical process like riming or aggregation (for example, dendritic crystals, observed by VDPS as oblate particles, which are falling though a supercooled liquid layer, are likely to transform into graupel, observed as isometric particles, due to a riming process). A new Section is inserted in the manuscript to illustrate this transition (Section 4.4). This shape transition informs that a microphysical process is occurring but it is impossible to give more details based on the VDPS method only. Further techniques are required, in addition, in such cases. Possible attempts (Doppler spectral separation, multi-wavelength, machine-learning based detection of liquid and graupel) are already mentioned in the manuscript.

(2) The Doppler spectral line of the particle type which contributes the strongest SNR in the co channel will be analyzed. Thus, as long as the same hydrometeor type is contained in the max SNR at all elevation angles, this one will be analyzed. In case of a mixture of columnar and dendritic crystals, SNR Doppler spectra will show two peaks at different fall velocities and SLDR will be only calculated for the main particle type which has the strongest signal in the co-channel. VDPS derives the particle shape or density for the predominant particle population in each layer. For this reason, it will likely not be the case that VDPS identifies aggregates due

to a mix of columnar and dendritic particles. At least as long only one particle type contributes to the strongest SNR at all elevation angles.

(3) Right, only the change in polarizability ratio might not be sufficient to distinguish pristine processes, aggregation, or riming. But just to see the evolution/change of the particle shape with cloud depth will help to interpret the scenario with the help of additional approaches (such as multi-wavelength, spectral, or machine-learning approaches such as VOODOO, as is also mentioned already in the manuscript).

(4) Finally, cloud-resolved spectral-bin models are capable to simulate aspect ratio and density of particles. We are currently working on transferring these model outputs into radar observations space. We are looking forward to the first model-obs evaluations which are scheduled for the abovementioned upcoming field experiment in Switzerland (winter 2024/25). At TROPOS, the modeling department is actually working with two different spectral-bin cloud-resolving models, COSMO-SPECS (Diehl et al., 2018) and AMPS (Hashino et al., 2020). Work is currently in progress to apply the Myagkov et al., (2016a) spheroidal scattering model as forward-simulator to the spectral-bin model output. This would then allow a model-obs-evaluation. We cannot mention these attempts yet in the manuscript, since work is ongoing an no references can be given so far.

(5) The reviewer is right that current models often assume a single habit throughout the temperature range. However, there are projects aiming to introduce a better habit prediction. One of them is a subproject of PROM (see section 2 in Silke Troemel 2021) driven by DWD and Uni Cologne. The project aims to implement a habit evolution throughout a cloud changing from a pristine shape at the top to aggregate and/or rimed particles. This habit prediction is planned to be incorporated into a Monte-Carlo Particle Model for Riming and Aggregation (McSnow, Brdar and Seifert, 2018). A dataset of profiles of polarizability ratios is of a great importance for evaluation of the habit evolution prediction. Also this approach is work in progress, which cannot yet be referenced.

**Comment 4 :**

Section 3 with the description of the methodology needs to be substantially reworked. I found the sentences and formulations (for example Sec 3.2) to be extremely confusing and hard to follow (see several specific comments below). I also found it disappointing that the retrieval code is not made publicly available to the community for example on github/gitlab. This should be a standard nowadays and would also help the community to further develop the approach.

**Response 4 :**

We are currently working on a publication of the retrieval code; the algorithm will be submitted after the date of submission of the revision. In the final version of the manuscript (upon acceptance), we will add a github-zenodo-reference to the final version of the code.

We worked heavily on Section 3 to make it more comprehensive for the reader. Main point is the removal of the cross-correlation coefficient from the method, which allows us to concentrate on the application of the SLDR. Beforehand, $\rho_{cx}$ was only required to deal with special cases where SLDR was rather low (<-25dB) while $\rho_{cx}$ still showed some elevation dependency. These cases used to be identified as slightly prolate particles. Now they are

grouped into the isometric particle class, which was already within the uncertainty range before, anyhow.

**III. Specific comments:**

**Comment 1 :**

L. 4: I find the formulation "wobbling effect of particles" unclear. I guess you want to say that SLDR is less sensitive to particle orientation as already shown by previous studies, correct?

**Response 1 :**

The reference to the wobbling effect is a common approach, as it has been already described in studies of Matrosov and Reinking (Reinking et al., 2002, Matrosov et al,.2012, 2020) and others.

**Comment 2 :**

L. 7: "SLDR-vs-elevation" maybe a bit unconventional. I suggest writing it out.

**Response 2 :**

We changed it line 8 to "SLDR from 90° to 30° elevation angle"

**Comment 3 :**

L. 9: "columnar and oblate" why not prolate and oblate or columnar and plate-like? This would be more consistent.

**Response 3 :**

We changed it to "columnar and plate-like", line 10.

**Comment 4 :**

L. 12-13: How can you be sure that the shape is not primarily changed by sublimation or depositional growth in different regimes (for example: first columnar and then plate-like resulting in capped-columns)?

**Response 4 :**

We do not make conclusions on the nature of the microphysical processes occurring in the investigated clouds. We are focusing on the vertical evolution of apparent particle shape only. What we get is the polarizability ratio which is the result of shape and density of the particle population. All further interpretation is subject to the reader, but will likely require further evaluation datasets (as explained in response II.3 and in the conclusions section of the article).

**Comment 5 :**

L. 19: I think a textbook reference would be more appropriate here for this first sentence.

**Response 5 :**

We add the reference to Pruppacher and Klett, 1997 to line 19 of the revised manuscript

**Comment 6 :**

L. 20: Add "the formation of" before precipitation.

**Response 6 :**

We added "the formation of" before "precipitation" to line 21.

**Comment 7 :**

L. 41-43: Another limitation of the spectral approach is that hydrometeors with similar terminal fall velocities (for example drizzle and small ice) cannot be distinguished in the spectrum. I suggest mentioning that several polarimetric variables can also be resolved spectrally. Would your method be principally applicable in a spectral approach?

**Response 7 :**

Indeed, analysis of the full Doppler spectrum would provide a much more detailed view on the particle populations present in a cloud layer. Actually, one project at TROPOS is currently dealing with such an approach, but an official publication is not yet available (see here for a conference abstract: https://www.spiedigitallibrary.org/conference-proceedings-of-spie/11859/2597693/Retrieval-of-shape-and-orientation-of-multiple-hydrometeor-types-from/10.1117/12.2597693.short?SSO=1 ). Analysis of Doppler spectra from different elevation angles is a challenge, as the horizontal wind and angular effects (caused by the changing antenna elevation angle) lead to a smearing of the Doppler peaks with increasing angular distance from the zenith.

However, SLDR in here is currently calculated for the main peak in SNR (co-channel) Doppler spectra representing the main population of particles. For example, the secondary ice production might be visible in the Doppler spectrum but is not taken into account in the method, as the overall co-channel SNR of secondary ice is usually below the SNR of the main hydrometeor population. We propose to add to lines 44-46 "*Moreover, hydrometeors with similar terminal fall velocities (for example drizzle and small ice) cannot be distinguished in the Doppler spectrum. In this case, it is possible to have a look at Doppler spectra of polarimetric parameters such as LDR or SLDR to confirm in which spectral mode crystals are present.*"

The discussion of Doppler-spectral approaches also motivated us to elaborate a bit more on the potential use of the co- and cross-SNR as reference spectra for selection of the main peak. Some text was added to lines 486-491 of the conclusions section: "*However, a new approach taking into account the comparison between main peaks detected in the co- and cross-channels can give more information about the ice crystal populations in a volume : if the main peaks are similar in the co- and cross-channels, it means that the main hydrometeor population depolarizes the most. On the other hand, the presence of different main peaks in the co- and cross-polarized Doppler spectra would imply the presence of a second hydrometeor population*

*which depolarizes strongly, while still a non-polarizing hydrometeor population dominates the co-channel signal".*

**Comment 8 :**

L. 56-58: That sentence contains some redundancy, or?

**Response 8 :**

Yes, Lines 57-60 were changed to : "*Note, that it is not directly possible to infer the aspect ratio and the apparent ice density from the polarizability ratio. However, since the polarizability ratio depends on both variables, we suggest using it to track the evolution of the ice particles from pristine state to aggregates and rimed particles.*"

**Comment 9 :**

L. 82: Maybe more general plate-like particles?

**Response 9 :**

Yes, we changed „dendritic" to „plate-like" particles at all instances in the revised manuscript.

**Comment 10 :**

Table 1: Please extend the table and add information on sensitivity at a certain range, beam width and co-cross-channel isolation.

**Response 10 :**

The co-cross-channel isolation was determined with the experimental approach described in Myagkov et al., 2015, by means of identification of the minimum SLDR that was measured in the presence of light drizzle. The co-cross-channel isolation was thus found to be -35dB. We added in the revised manuscript this sentence lines 133-135: "*The co-cross-channel isolation was determined with the experimental approach described in Myagkov2015, by means of identification of the minimum SLDR value that was measured at zenith-pointing, in the presence of light drizzle. The co-cross-channel isolation was thus found to be -35 dB with MIRA-35.*".

**Comment 11 :**

L. 95: figure axes instead of caption?

**Response 11 :**

L.107 we changed „*throughout this article and on the figure captions*" to "*throughout this article and figures.*"

**Question 12 :**

L. 115/Eq(2): Explain meaning of <>

**Response 12 :**

We added L.125 : "*...and <> denotes averaging over a series of collected Doppler spectra.*"

**Comment 13 :**

L. 116: "subject to noise artifacts". Unclear what you mean here. I thought only the maximum peak is used…

**Response 13 :**

Yes, it is right, we don't correct SLDR spectra but SNR spectra. We changed line 126 to "*The raw spectra of SNR are subject to noise artifacts.*"

**Comment 14 :**

L. 150: "gradients of rho_s against the elevation angle" What do you mean here? I can't see a gradient against an elevation angle as you only show 90° and 150°.

**Response 14 :**

Yes, you are right we don't measure any gradient but a slope. We changed „*gradient*" to "*linear regression*" at all instances the revised manuscript. Concerning Figures 2 and 3, the distribution of modeled SLDR is plotted in Figure 3 for data points defined by the respective symbols in Figure 2. The scattering model is actually processed for all elevation angles and for each elevation step a value associated to the respective particle shape class is plotted in Figure 3.

**Comment 15 :**

L. 151: "giving an idea about the shape" please rephrase.

**Response 15 :**

We have changed the following sentence in lines 166-168 : "*For our radar configuration, the realistic range of possible polarizability ratios $\xi$ spans from 0.3 to 2.3 giving an idea about the shape of particles, while the degree of orientation $\kappa$, ranging from -1 to 1, describes the orientation of particles.*" to "*For our radar configuration, the realistic range of possible polarizability ratios $\xi$ spans from 0.3 to 2.3 and the degree of orientation $\kappa$, is ranging from -1 to 1.*".

**Comment 16 :**

L. 143: "assumes Rayleigh scattering" I suggest adding some sentences in the discussion that the Rayleigh assumption of the current technique should be checked with more realistic scattering methods. For example, Schrom and Kumjian, JAMC, 2018 show that typical radar polarimetric variables of plate-like particles calculated with T-Matrix deviate quite substantially from DDA calculations even at low weather radar frequencies. In my opinion, those results should be a warning that similar uncertainties might also affect SLDR methods. Ideally, this manuscript should provide a consistency check with existing DDA scattering databases.

**Response 16 :**

Please see our reply to the comment I.3 on the Rayleigh scattering assumption.

There is a number of advantages the simple spheroidal model provides unlike DDA:

1. It considers the volumetric scattering
2. It can explain general polarimetric signatures observed in natural clouds
3. It has just a few parameters
4. These parameters are well constrained by the measurements
5. The optimization-based retrieval is quick

We added these information to Section 2.3 of the revised manuscript lines 154-163.

**Comment 17 :**

Fig. 2: If find it quite confusing to denote a "domain" by single symbols. Why do you not include some dashed lines separating the different domains and add some letters A, B, C for referencing to those regions? Add delta= or Elevation= on the top of the columns in front of the number.

**Response 17 :**

On this Figure we show exactly which values we took from the full matrix of possible solutions of the spheroidal model. But we agree that the word "Domain" is misleading. We changed the following sentence in the caption of Figure 2 : "*'+' (oblate particles), 'O' (isometric particles) and '\*' (prolate particles) symbols are data points used in Figure 3. The elevation dependency of these three scenarios are depicted further in Section 2.3. The two white vertical lines in (a) separate the three particle domains of oblate (zone A), prolate (zone B) and the isometric (zone C) hydrometeors.*"

**Comment 18 :**

L. 187: I see the point that only using two elevation angles reduces the total time needed for a scan. But on the other hand, aren't you also losing information by reducing the number of elevation angles so drastically? Can you provide an evaluation of what is the best compromise between scan speed and number of elevation angles included?

**Response 18 :**

All elevation angles of an RHI scan are processed to calculate SLDR($\theta$min) and SLDR($\theta$max) by means of the 3rd order polynomial fit (Section 3.1) and to calculate the slope of SLDR vs. elevation by means of the linear regression (Section 3.3). Only SLDR($\theta$min) and SLDR($\theta$max) are needed for the evaluation against the output of the spheroidal scattering model (Section 3.2). For more details, see our reply to comment 1. Section 3 was improved during the review process. We think it will now be easier for the reviewer and the reader to understand the methodology.

We cannot provide this kind of evaluation about the best compromise between scan speed and number of elevation angles, which depends on various settings of the cloud radar (e.g., scan

speed, sensitivity, co-cross-channel isolation). A limit number of data points is already given in the manuscript (15%).

**Comment 19 :**

L. 190: I agree that the need to assume horizontal homogeneity is a very strong assumption limiting the applicability of the retrieval. But how broadly applicable is your retrieval method if one needs an "experienced-eye evaluation" of each RHI scan? I think this is a big limitation and you should better provide some more objective criteria which can be used.

**Response 19 :**

This question has been already answered in response I.4. We added a sentence to the Section 3 of the revised manuscript.

**Comment 20 :**

L. 199: What do you mean with "scan pattern". Fig. 2 only shows two constant elevation angles, right? Or do you mean Fig. 1?

**Response 20 :**

Yes, we confused the Figure number. We have corrected in Section 3.1, lie 221 "Figure 2" to "Figure 1".

**Comment 21 :**

L. 217: I probably don't understand the method correctly: I thought you measure at 90° and 150° elevation only (L. 187). What I don't get is why you need to apply a polynomial fit to obtain the values at those angles? Also how can you apply a polynomial fit if you have only two points? Or are you using complete RHI scans to derive the fit which you can then apply to only two-elevation angle measurements? Please explain better.

**Response 21 :**

Yes, we are using complete RHI scans to calculate $SLDR(\theta_{min})$ and $SLDR(\theta_{max})$ via the $3^{rd}$ order polynomial fit and the linear regression of SLDR for obtaining the general slope of the SLDR vs. elevation angle (see replies to questions 1 and 3 above). The polynomial fit is applied to all data points of the complete RHI scan for each height level and thus acts as a kind of noise filter. The retrieval calculates $\theta_{min}$ and $\theta_{max}$ (minimum and maximum elevation angles of the RHI scan) already mentioned in Section 2.3. We propose to rephrase the sentence lines 206-207 to *"has the particularity of combining simulated and measured values of SLDR at only two elevation angles, isolated from a full RHI scan."* and suppress *"of 90° and 150°"*.

**Comment 22 :**

L. 221: Unclear what you mean. In Fig. 2a I see that delta_s is very "straight" for prolate and isometric particles. For the oblate ones, I see that a non-linear fit is needed. Please rephrase. Again in the entire paragraph it is unclear to me what is fitted against what.

**Response 22 :**

We rephrased the second part of Section 2.3 (lines 149-203) to be more coherent with Figure 2a.

**Comment 23 :**

L. 231: Hard to follow without references. Is this valid for all cloud radars even at all different wavelengths? What are typical minimum values of delta_s of common cloud radar systems? The references in Sec. 2.1 are only valid for your MIRA system, or? How low was delta_s for the systems used in previous studies, for example the Ka and W band systems used in the US?

**Response 23 :**

This question is already answered in response III.10. We removed the reference because the SLDR limit value of about -31dB was valid only for the radar calibration of LDR-mode MIRA-35 as used by Bühl et al. (2016). The minimum value of SLDR (co-cross-channel isolation) used in our study was derived as proposed by Myagkov et al., 2015, by identifying the minimum detected SLDR in light rain.

**Comment 24 :**

L. 233: Confusing. In Fig. 2c, d the minimum (dark blue) values of the colorbar are indicating -35 dB.

**Response 24 :**

Already answered in the previous question.

**Comment 25 :**

L. 241: What is the prediction interval of a polynomial fit? Please explain better.

**Response 25 :**

We added the following sentence to lines 268-271 : "*The confidence interval is calculated as follows : Δ95 =2Δ where Δ is the standard deviation of the difference between the measured and simulated values of SLDR at all available elevation angles from 90° to 150°.*". The Section 3.1 has been completely rephrased to be better understandable for the reader.

**Comment 26 :**

Section 3.2 and Caption of Fig.5: In the second part of section 3.2 and especially in the caption to Fig. 5 I got completely lost. I tried to read and understand it now several times but I am totally confused. For example some formulations of caption to Fig. 5: " intersection of observed delta_s in the data fields simulated with the spheroidal scattering model". How can observed

delta_s be simulated? "Isolines of the observed values in the model space" Isolines of observed values? "Overlay of intersections of the isolines" what is that? "As observational input, hypothetical values of typical oblate particles…" What is an observational input? I think the section and the caption need to be carefully rephrased. Please give it to somebody who is not an expert in order to ensure that an average experienced reader can follow.

**Response 26 :**

Thanks for the comment and apologies for the complicated formulations in the original version of the manuscript. We rephrased and reorganized the methodology section and the caption of Figure 5 in order to make it more accessible to the reader (see Section 3).

**Comment 27 :**

L. 275: What is a "hypothetical case study" and what should it be useful for??

**Response 27 :**

The hypothetical case study is useful to better illustrate the VDPS method. Values of SLDR are chosen to represent plate-like crystals and will be used to depict the method step by step. As we don't have real evaluation data, we needed to select a hypothetical case study to illustrate the application of the VDPS technique for a certain, well-defined hydrometeor type. We rephrased the hypothetical case study in an example and incorporated this case study better in the text to describe the method (See Section 3 of the revised manuscript).

**Comment 28 :**

L. 281: "Stratification of ice particle shapes" again unclear what you mean by that. Maybe a vertical profile?

**Response 28 :**

Yes, we changed lines 311-312: "*Stratification of ice particle shapes*" to "*vertical profile*".

**Comment 29 :**

Figure 7: The temperature information used from model analysis in your Cloudnet classification has obviously some problems as the melting level jumps unrealistically up and down over time (for example between 13:00-13:30 and also later on). How does that affect your results? You should check the model fields or only rely on the melting layer estimated from the radar itself.

**Response 29 :**

The temperature is not responsible for the sudden decrease of the melting layer height. If you have a look at Figure 8, the melting layer is well represented at around 1300m height (high SLDR values), which corresponds to the melting layer detected by Cloudnet at 13:00 UTC. The drop of the melting layer to 1000m height results from the melting layer detection scheme of Cloudnet but the temperature range represented in Figure 7 by black dashed lines is realistic. Please note, in the shown case study, Cloudnet identified the melting layer during the time of

the scan at a wrong height level. Reason for this is, that Cloudnet uses the observed LDR, fall velocity and modeled dew point temperature as indicators for the melting layer. During the discussed period, the SLDR and fall-velocity signature of the melting layer was too weak, which caused Cloudnet to switch to the 0°C-dew point level for the melting layer classification. The actual location of the melting layer is further highlighted in Figure 8, as will be discussed in the remainder of this subsection. This problem doesn't affect our results because we only use the polarimetric parameter SLDR (Cloudnet is shown to put the scan into the context of the overall meteorological situation) and we validate the method predominantly based on the temperature which presents no problems in this case.

We added in Section 4.1, lines 344-347 : *"The sudden drop of the melting layer height from 1300 m to around 1000 m height that is visible right at the time of the RHI scan, is an artifact of the melting layer detection scheme of Cloudnet, which switched from a fall-velocity-based detection to the 0°C-dewpoint level as threshold for the melting layer identification.."*

**Comment 30 :**

Section 4.1: It is still unclear to me how you can use modeled values of delta and rho based on pure ice particles for this rain case. In the previous chapter, you argue that zeta is "not strongly affected by the refractive index" but now you are using observed delta and rho values as well? By the way, as in most of the manuscript you are using an unscientific formulation "not strongly affected by the refractive index". You should quantify what you mean by that: Less than 10%, less than 1% or something similar. And what is actually the problem of using refractive index of water for this case study? I guess the Rayleigh scattering parameters won't take much time to be calculated.

**Response 30 :**

Refractive index does not matter in the calculation of polarimetric properties of isometric particles. polarizability ratio is a mix of aspect ratio and refractive index. We did not select the ice phase for the isometric case study, as we cannot be sure about the actual ice particle shape. For droplets in light rain, we can be sure that they are isometric.

**Comment 31 :**

Description of Fig. 9: In the plot you show linear fits which have by definition a constant gradient (derivative is the slope). If you want to indicate that the gradient changes over the range of elevation angles then you should add a curve that represents a derivative over a certain elevation angle range. Right now I don't find this very consistent and physically sound.

**Response 31 :**

You are right, we don't use a gradient but the slope of the linear fit to describe the behavior of SLDR measurements from 90°($\theta_{min}$) to 150°($\theta_{max)}$ elevation angle. We changed "gradient" to "linear regression" in all manuscript (already answered in reponse III.14).

**Comment 32 :**

L. 327: At 1300m I find in the blue box area of Fig. 7 temperatures far below 0°C. But again I think this is an artifact of the model temperature field used for Cloudnet.

**Response 32 :**

The 0°C isoline is at around 1400m height which means that the temperature is positive at 1300m height (and not below 0°C). If you mean that the 0°C isoline and melting layer are not correlated, it is assumed that ice particles melt at the level of 0°C dewpoint temperature, which is usually at heights below the 0°C temperature height level. Concerning the melting layer jump, we consider the melting layer visible in Figure 8 was detected at around 1300m height.

**Comment 33 :**

L. 330-339: Sorry, but this interpretation of particle shape above the melting layer is completely invalid in my opinion. Your method implicitly assumes that the volume is composed of only one particular particle shape, correct? In such a relatively thick mixed-phase cloud it is almost certain that close to the ML you face a mixture of single crystals, aggregates and maybe even rimed particles. This mixture is not only comprised of very different shapes but also various densities.

**Response 33 :**

Yes. In our study we calculated SLDR using only one spectral line in the co-channel which indicates the dominant type of particles. We only mentioned that particles are isometric or less dense (as assumed, using the main SNR peak), which can be formed via processes such as riming (graupel are spherical) or aggregation (less dense particles). We only describe the polarizability associated in this layer and we don't give details about the microphysical process responsible in this case.

We changed lines 369-371 : ", *as both of which form isometric graupel particles or aggregates, respectively*" to "*…it is likely that these isometric or less dense particles are the result of mixed-phase cloud processes, such as riming or aggregation, which cannot unambiguously be identified solely with the VDPS method.*".

**Comment 34 :**

L. 367: It is not true that between -10 and 0°C one expects only columnar particles. For example, between -8 and -10°C one expects plate-like particles.

**Response 34 :**

Thank you for this comment. For this reason, we changed the columnar case study, that is shown in Section 4.2. The new case study features a temperature range from -3°C to -7 °C which allows the formation of columnar crystals only.

**Comment 35 :**

L. 382-383: "rather dense" and dendritic particles can't be true. Due to the branching of dendrites their density is much less than for example hexagonal plates.

**Response 35 :**

You are right, we can be more precise, we changed the interpretation and describe hydrometeors as thick plates, which is more consistent with our results. We propose to change in Section 4.3, lines 413-415, "*The values of ξ are relatively constant around 0.4 from 3100 m to 3600 m height corresponding to particles which are strongly oblate and rather dense, pointing likely to the class of thick plate crystals (Reinking et al., (2002), Matrosov et al., (2012). On the other hand, above 3600 m height, ξ≈0.55 was observed, representing particles which are likely less dense such as plates or dendritic crystals.*"

**Comment 36 :**

Fig. 16: The distribution of rho_s does not look like following a linear relationship. What does this indicate and why not using also polynomial fit for rho as well?

**Response 36 :**

The $3^{rd}$ order polynomial fit is only used to calculate $SLDR(\theta_{min})$ and $SLDR(\theta_{max})$ for comparison of these values to the respective output of the spheroidal scattering model at the min and max elevation angles $\theta_{min}$ and $\theta_{max}$. However, $\rho_{cx}$ was not used in the spheroidal scattering model to calculate the polarizability ratio (only SLDR is used) and is removed completely from the new version of the revised manuscript (explained in more details in response I.1).

**Comment 37 :**

L. 415: You claim that you developed an automatized framework but in L. 191 you say that the method needs "experienced-eye evaluation of the obtained RHI scan". How can this method be automatized if you need to inspect each single RHI scan?

**Response 37 :**

The framework works automatically but doesn't select the RHI scan. We find the "experienced-eye evaluation" more relevant to validate the homogeneity of studied clouds." However we can consider developing a new algorithm that is able to select RHI scans according to relevant parameters such as fall velocity of particles by using Doppler spectra from 90° to 150° elevation angle. We removed the following sentence from the conclusion Section of the original manuscript: "*The VDPS method was implemented by means of an automatized framework, which permits us to obtain statistics about the particle shape for a long period of measurements and covering several field campaigns.*"

**References:**

Brdar, S., & Seifert, A. (2018). McSnow: A Monte-Carlo particle model for riming and aggregation of ice particles in a multidimensional microphysical phase space. *Journal of Advances in Modeling Earth Systems*, 10, 187– 206. https://doi.org/10.1002/2017MS001167

Bringi, V. and Chandrasekar, V.: Polarimetric Doppler Weather Radar: Principles and Applications, https://doi.org/10.1017/CBO9780511541094, 2001.

Bühl, J., Seifert, P., Myagkov, A., and Ansmann, A.: Measuring ice- and liquid-water properties in mixed-phase cloud layers at the Leipzig Cloudnet station, Atmospheric Chemistry and Physics, 16, 10 609–10 620, https://doi.org/10.5194/acp-16-10609-2016, 2016.

Diehl, K. and Grützun, V.: Model simulations with COSMO-SPECS: impact of heterogeneous freezing modes and ice nucleating particle types on ice formation and precipitation in a deep convective cloud, Atmos. Chem. Phys., 18, 3619–3639, https://doi.org/10.5194/acp-18-3619-2018, 2018.

Hashino, T., G. de Boer, H. Okamoto, and G. J. Tripoli, 2020: Relationships between Immersion Freezing and Crystal Habit for Arctic Mixed-Phase Clouds—A Numerical Study. *J. Atmos. Sci.*, **77**, 2411–2438, https://doi.org/10.1175/JAS-D-20-0078.1.

Matrosov, S. Y., Reinking, R. F., Kropfli, R. A., Martner, B. E., and Bartram, B. W.: On the Use of Radar Depolarization Ratios for Estimating Shapes of Ice Hydrometeors in Winter Clouds, Journal of Applied Meteorology, 40, 479 – 490, https://doi.org/10.1175/1520-0450(2001)040<0479:OTUORD>2.0.CO;2, 2001.

Matrosov, S., Mace, G., Marchand, R., Shupe, M., Hallar, A., and McCubbin, I.: Observations of Ice Crystal Habits with a Scanning Polarimetric W-Band Radar at Slant Linear Depolarization Ratio Mode, Journal of Atmospheric and Oceanic Technology, 29, 989–1008, https://doi.org/10.1175/JTECH-D-11-00131.1, 2012

Myagkov, A., Seifert, P., Wandinger, U., Bauer-Pfundstein, M., and Matrosov, S. Y.: Effects of Antenna Patterns on Cloud Radar Polarimetric Measurements, Journal of Atmospheric and Oceanic Technology, 32, 1813–1828, https://doi.org/10.1175/JTECH-D-15-0045.1, 2015.

Myagkov, A., Seifert, P., Bauer-Pfundstein, M., and Wandinger, U.: Cloud radar with hybrid mode towards estimation of shape and orientation of ice crystals, Atmospheric Measurement Techniques, 9, 469–489, https://doi.org/10.5194/amt-9-469-2016, 2016a.

Myagkov, A., Seifert, P., Wandinger, U., Bühl, J., and Engelmann, R.: Relationship between temperature and apparent shape of pristine ice crystals derived from polarimetric cloud radar observations during the ACCEPT campaign, Atmospheric Measurement Techniques, 9, 3739–3754, https://doi.org/10.5194/amt-9-3739-2016, 2016b.

Reinking, R. F., Matrosov, S. Y., Kropfli, R. A., and Bartram, B. W.: Evaluation of a 45° Slant Quasi-Linear Radar Polarization State for Distinguishing Drizzle Droplets, Pristine Ice Crystals, and Less Regular Ice Particles, Journal of Atmospheric and Oceanic Technology, 19, 296 – 321, https://doi.org/https://doi.org/10.1175/1520-0426-19.3.296, 2002.

Ryzhkov, A. V.: Interpretation of Polarimetric Radar Covariance Matrix for Meteorological Scatterers: Theoretical Analysis, Journal of Atmospheric and Oceanic Technology, 18, 315 – 328, https://doi.org/10.1175/1520-0426(2001)018<0315:IOPRCM>2.0.CO;2, 2001

Schrom, R. S., and M. R. Kumjian, 2018: Bulk-Density Representations of Branched Planar Ice Crystals: Errors in the Polarimetric Radar Variables. J. Appl. Meteor. Climatol., 57, 333–346, https://doi.org/10.1175/JAMC-D-17-0114.1.

Westbrook, C. D., and Illingworth, A. J. (2011), Evidence that ice forms primarily in supercooled liquid clouds at temperatures > −27°C, *Geophys. Res. Lett.*, 38, L14808, doi:10.1029/2011GL048021.

---

## Author Response (AR2)

**Response to the Editor**

Dear Editor,

We once more want to thank the reviewers for taking the time for a second review and the provision of new valuable comments. Comments and suggestions presented by both of reviewers 1 and 2 have been considered in the new version of the manuscript.

Specifically, we took note of the concerns both reviewers raised again about the application of the Rayleigh scattering theory in the VDPS retrieval. We provide further explanation for our decisions below in our replies to the specific review comments (see R1-C1, R1-C2, R2-C5, R2-C6 below). We decided to put some general statements already here. First, we'd like to emphasize once more that we introduce the VDPS method as a versatile, additional tool for the characterization of the microphysical structure of mixed-phase cloud systems, which can be applied to data from meanwhile widely available scanning SLDR cloud radars. This is highlighted already in the introduction section of the manuscript. Similar to many other radar observables (such as even the radar reflectivity factor Z), it relies on certain assumptions (such as the assumption that Z is always given with respect to liquid water, even in ice clouds). Second, the sole application of SLDR for estimating the polarizability ratio (as an indicator for the shape) of hydrometeors relies on the availability of a simple model. Much more complex measurements (multi-frequency, spectrally resolved, symmetrically scanning observations of radars with very accurate reflectivity calibration) would be needed in order to constrain the degrees of freedom introduced by complex T-Matrix or DDA scattering models, as these models depend strongly on assumptions on size and specific shape of the scatters. When a wrong size distribution is assumed in DDA or T-Matrix calculations, resonance effects can lead to strongly varying simulations of backscattering and polarimetric results. Third, we inform and elaborate in detail about the possible deficiency of the Rayleigh-scattering assumption in our manuscript. By doing so, we provide the expertized reader with information about possible uncertainties in specific cloud situations while hoping to provide motivation for future improvements of the approach, which is beyond the current state of the VDPS method as it is presented in our study.

Additionally, we would like to complete our answer to one comment of the second reviewer from the first revision cycle, which was "*Can you explain better how you think a vertical profile of shape can be useful for model development? I am very skeptical since most bulk schemes assume a constant ice particle habit throughout the entire temperature range. Only very few experimental models currently exist which take particle habit into account.*" We would like to refer to a recent manuscript well submitted to Journal of Advances in Modeling Earth Systems by the German Weather Service (preprint available at DOI:10.22541/essoar.168614461.18006193/v1). In the manuscript the authors use the polarizability ratio to improve the ice growth characterization for the explicit habit prediction in the Lagrangian super-particle ice microphysics model McSnow. We added this information to lines 58-64 of the introduction section of the updated version of our manuscript: "*However, since the polarizability ratio depends on both variables, it can be used to track the evolution of the ice particles from pristine state to aggregates and rimed particles in observational studies. Polarizability ratio profiles are also valuable for modeling studies since the profiles can be used to constrain microphysical processes of ice growth. The first attempt to utilize polarizability ratios to improve ice characterization in models was recently done by Welss et al. (2023). Based on polarizability ratios the authors have updated the ice growth characterization for the explicit habit prediction in the the Lagrangian super-particle ice microphysics model Mc-Snow developed by German Weather Service (DWD, Brdar and Seifert (2018)).*"

Consequently, we would like to submit the revised manuscript and the diff-version of the revised manuscript together with our responses to all the comments provided by the two reviewers. In our replies, all references to modified lines are given with respect to the final diff-version of the manuscript (with differences) to facilitate the lecture to the editor and/or reviewers. Comments are enumerated by the term "Reviewer_1/2_Comment_X which means e.g. R1-C1". In the corresponding answers, the C ("comment") is exchanged by an A ("answer"). Thank you for considering our work.

Best regards,

Audrey Teisseire, Patric Seifert, Alexander Myagkov, Martin Radenz and Johannes Bühl.
teisseire@tropos.de

**Response to reviewer 1# :**

R1-C1: My question on Rayleigh scattering is yet addressed... I am concerned with the non-Rayleigh scattering effect in presence of large aggregates at Ka band, but you assume Rayleigh at all conditions. I am questioning the applicability of this assumption.

R1-A1: We acknowledge the concerns raised by reviewer 1. But we would also like to note that we extensively discussed the possible effects of non-Rayleigh scattering in our reply to reviewer 1 and we already added a comprehensive discussion to section 2.3 of the 1st revision of the manuscript. Nonetheless, we now also elaborate on the potential shortcomings of the Rayleigh approximation in the conclusions section of the manuscript. We added to lines 511-515: "*Finally, it remains subject of future discussions to investigate the applicability of more sophisticated scattering theory in a quantitative determination of hydrometeor shape. The scattering model underlying the VDPS method only requires information about the axis ratio, apparent density and canting angle distribution. T-Matrix or DDA methods provide much more degrees of freedom concerning the microphysics of the scattering hydrometeors, specifically the number size distribution and fine structure of the hydrometeors*"

Unlike the VDPS method which requires only two parameters (namely polarizability ratio and degree of orientation), T-matrix and DDA require additional characterization of particles. In the case of T-matrix, size is required. Depending on the exact size-wavelength relations, scattering properties of a single particle can indeed deviate from the Rayleigh solution due to resonance effects. However, in the radar scattering volume we have a huge number of particles with different sizes and therefore, resonance effects are averaged out. In the case of DDA, the exact geometric structure of a particle must be characterized. In this case as well, there can be scattering properties deviating from Rayleigh, but again here due to a number of particles with different sizes and shapes the resonance effects are averaged out. In addition, these additional degrees (size, fine geometric structure) of freedom cannot be constrained by observations at only one wavelength. A combined analysis of the radar polarimetry and multi-frequency observations might add such a constraint, but at the moment even though such datasets exist (TRIPEX-pol von Terzi et al., 2022), they are not widely available (Kneifel et al.,2011, Leinonen et al., 2013).

R1-C2: In the response, comments 5 & 7 were poorly addressed.

Comment 5:

I believe that the Rayleigh condition at Ka-band should be considered. The theoretical basis is based on the assumption of Ka-band Rayleigh scattering which is satisfied for pristine ice. However, in presence of large aggregates which usually occur at -5 to 0 C or -15 C, this assumption is violated and the retrieval should be made with caution.

Comment 7 :

The authors claim that rimed or aggregated ice particles are isometric and speculate that the retrieved polarization ratio around 1 indicates riming or aggregation. I feel that this statement should be used with caution. Firstly, the method may be limited to pristine ice at ka band. The Rayleigh condition at Ka band may not have reached, for example at 1.6 km in figure 10. Therefore, the basic assumption is broken. Secondly, very heavily rimed ice can be isometric. Lightly rimed ice or aggregation has a characteristic aspect ratio.

R1-A2: The proposed method is applied to spectrally resolved polarimetric measurements. This way we reduce the likelihood of having considerably different particle types affecting the

measurements. When we have pristine ice particles, VDPS is sensitive to particle's shape and density. When we have aggregated/rimed particles, polarimetric signatures are weak, which indicates that these particles are either isometric or only slightly non-spherical or have low density (due to air inclusions) or both. In order to illustrate this, we provided measurement statistics in our previous reply. Here we show it again in Figure 1 representing a single scatter plot of 80 deep precipitating clouds randomly chosen from the analyzed campaign. For each cloud we took profiles of the attenuated reflectivity and SLDR, at 60° off-zenith angle. Small particles (Rayleigh scattering, low reflectivity) are in the left part of the plot. These particles often produce strong polarimetric signatures (large SLDR up to -5 dB). In the right part of the plot we have big particles. We do not know the type of these big particles but they produce SLDR mostly below -10 dB which is equivalent to 0.9 dB ZDR. When these values of SLDR (< -10 dB) are used for VDPS, the resulting polarizability ratio is in the range of 0.8 and 1.2, Values in this range indicate that particles are isometric (see Table 2), i.e. do not produce strong polarimetric signatures. And this is the correct result. We do not see big violations of the retrieval applied to these big particles as suggested by the reviewer.

Concerning the prolate particles detected at 1.6 km in Figure 10 of the first version of the manuscript, there were artifacts caused by the calculation of SNR based initially on the spectral line in the cross channel, and increasing values of the cross-correlation coefficient describing irregular shapes but not especially columnar crystals. In the revised version of the manuscript, we can see that VDPS derives isometric particles at this layer.

[Figure]

*Figure 1: Attenuated radar reflectivity Z and SLDR of 80 deep precipitating clouds as observed at 60° off-zenith angle in the course the CyCARE campaign.*

R1-C3: Discussion on the weakness of this work is missing in conclusions. For example, no direct validation with in-situ observations

R1-A3: Thanks for highlighting, that the discussion seems to be still missing to elaborate on the weaknesses of the study. We added that so far validation with in-situ observations was not possible. While we pointed to this issue already in the reply to the first review, we missed to include it into the actual manuscript. We now did so, see lines 485-487: "*It's important to*

*highlight that we could not validate the method using in-situ observations throughout the two campaigns. It is nevertheless the goal of the authors of this study to aim on deployments of the SLDR-mode scanning cloud radar in campaigns where in-situ observations are available.*"

R1-C4: A fatal error in response 9 and 10. Spectral broadending will definitely affect the values of spectral peaks at both channels, since the peak value is affected by adjacent data during the convolution, namely smoothing. Therefore, your SLDR will be affected. You could simply follow Kollias 2011 and do forward simulation.

R1-A4: We apologize for the imprecise reply. What we meant is, that a convolution of several spectral lines with the same polarimetric signatures does not change these polarimetric signatures. For instance, at a cloud top where a high turbulence is expected, we have likely pristine ice particles with the same shape. These particles produce similar polarimetric signatures and these signatures are still the same in the case of broadening.

When we analyze a volume in the middle of a deep precipitating cloud, the maximum spectral line typically corresponds to large aggregated/rimed particles. Here again neighboring spectral lines characterize similar types of particles. And therefore, the spectral broadening would not considerably change observed polarimetric signatures.

The broadening would only have an effect when particles with different polarimetric signatures are "mixed/convolved" into a single spectral line. But in this case, we argue that most of the time particles with different polarimetric signatures have considerably different terminal velocities and this difference is too large for turbulence to mix these particle populations into the same spectral line.

However, in order to acknowledge the possible impact of spectral broadening on the polarimetric signatures, we added a text passage to lines 505-511 of the conclusions section, which discusses the potential of spectral broadening and provides the reference to the study of Kollias et al, 2011: *"In addition, turbulence, horizontal wind, and radar beam width, especially at large off-zenith pointing angles, can lead to a broadening of the Doppler spectra, which has the potential to impact the spectral peak values in both channels (Kollias et al., 2011). Spectral broadening becomes noteworthy when particles with distinct polarimetric signatures are blended into a single spectral line, and it becomes particularly relevant when substantial turbulence is present (typically on the order of several meters per second). However, the spectral broadening would not considerably change observed polarimetric signatures in the case of pristine ice crystals at the cloud top, or when only one type of hydrometeors is present in a cloud volume. "*

R1-C5: You should give specific definition for isometric particles, or use other descriptors, since isometric is rarely used in cloud physics communication.

R1-A5: We added an additional description of the primary shape classes at lines 171-177 of the manuscript: *"In this study, we will sort particles into three primary categories based on their shape: Oblate particles, which have a polarizability ratio less than one, prolate particles, characterized by a polarizability ratio greater than one, and isometric particles, where the polarizability ratio is ranged from 0.8 to 1.2, depending on the radar calibration (see Table 2). With respect to the definition in this study, we consider particles as isometric when they do not produce considerable polarimetric signatures. Such particles have either spherical or just slightly-non-spherical shape. In addition, non-spherical particles with low density (low-refractive index) also appear to be isometric. "*

**Response to reviewer 2# :**

I would like to thank the authors for carefully revising the manuscript. In my opinion it has greatly improved and I find it now much easier to follow. I think the manuscript is now ready to be published after some minor comments and changes listed below are taken into account.

General comments:

R2-C1: One advantage of the method which the authors might consider is that the SLDR method is probably relatively immune to attenuation effects. For ZDR one has to consider effects of differential attenuation. RHI scans at multiple frequencies suffer from the need to estimate the dual-frequency relative attenuation along the slant path. I guess this is all much less relevant for SLDR.

R2-A1: Note, that in the STSR mode (simultaneous transmission and simultaneous reception) in which ZDR is measured, the transmitted signal is exactly the same as in the SLDR mode in terms of polarization, i.e., linearly polarized at 45 degrees electromagnetic wave is transmitted. Effectively, such a wave is a superposition of two waves of which one is horizontally polarized and one is vertically polarized with the 0°- phase shift.

Therefore, in both STSR and SLDR modes, the signal is transformed in the atmosphere in the same manner by scattering and propagation effects. The difference between STSR and SLDR modes is only the basis in which the signal is received, but in both modes the received signal is still affected by the atmosphere in the same way. It could be that partially some of these effects are mitigated in SLDR but this would require investigations which are beyond the focus of the current manuscript.

R2-C2: The four case studies are quite interesting and demonstrate the potential of the method. But honestly, I would not consider them as a true evaluation of the method. I agree that they are very valuable consistency checks but without in-situ truth, I think it is not a real evaluation.

R2-A2: Indeed, with the statement 'evaluation', we did not mean to do an actual validation but more an application. We thus implemented two modifications into the new version of the manuscript.

We rephrased Lines 480-485 to: "*A fourth case study demonstrated the potential of the VDPS method for tracking of the evolution of the ice crystal shape between top and base of a deep cloud system. Before application of the VDPS method to the case studies, the algorithm was tested and calibrated with success based on observational datasets from two field campaigns...*"

We added in addition another statement on the lack of actual evaluation data to Lines 485-487: "*It is important to highlight that we could not validate the method using in-situ observations throughout the two campaigns. It is nevertheless the goal of the authors of this study to aim on deployments of the scanning SLDR radar in campaigns where in-situ observations are available.*"

Specific comments:

R2-C3: Table 1: The sensitivity of a radar is range dependent. Please provide range and averaging time used to estimate the sensitivity.

R2-C3: We now provide the averaging time (1s) and range (5 km) in Table 1.

R2-C4: L. 106: "receiver antenna". Why specifically receiver antenna, I thought the same antenna is used for RX and TX? As such also the TX polarization is turned by 45°, or?

R2-C4: Indeed, apologies for the typo. We removed the term 'receiver' from line 103.

R2-C5: Discussion of scattering in section 2.3: I thank the authors that they discuss the potential issue of non-Rayleigh scattering for this method. I still think the paper would be much stronger if some sample comparisons of polarizability ratios from idealized plates and columns would be compared from the standard Rayleigh spheroidal method and some existing DDA simulations (for example in a short supplement). However, I suggest to add some example references to those statements "It is well known that the Rayleigh approximation is not always applicable to simulate scattering from large ice particles. Often the direct dipole approximation (DDA) is used to simulate scattering of individual ice particles having a complex shape." I suggest especially mentioning studies looking at non-Rayleigh scattering effects of pristine ice particles and scattering databases that contain the full scattering information of single ice crystals.

R2-C5: A similar question was already raised by reviewer 1 in R1-C1, that is why we kindly refer to our response R1-A1.

As requested by reviewer 2, we did an additional literature survey and added further references to Section 2.3 of the revised manuscript, lines 159-167. We now cite the initial introduction into DDA simulations (Draine and Flatau, 1994, line 161), refer to one of the most-extensive currently existing scattering database (Lu et al., 2016, line 162) and also give several further references to studies which used DDA simulations and what they found about the backscattering and polarimetric signatures of hydrometeor populations.

R2-C6: L. 158: "In general, ice particles in a scattering volume have arbitrary shapes and the contribution of individual particles to the backscattering radar observables is averaged out." I don't understand why this is an argument of not using DDA particles. Also in your method, you assume that you have only one particle type in your volume.

A2-A6: In order to make a simulation of an individual particle with DDA one needs to characterize the fine geometric structure of the particle. In this case the scattering, which in case of large size is defined by resonance effect, strongly depends on the exact shape of the particle and its orientation relative to the radar. In reality there are many particles in a scattering volume and their shape is not known apriori. Since resonance effects on individual particles depend on their size, shape, and orientation and these parameters are likely different for all particles in the volume, the observed polarimetric signatures can hardly be represented by DDA scattering of an individual particle. One needs to at least simulate plenty of different particles and average their scattering properties. Ultimately, one would have to presume the distribution of particle fine structure over the entire hydrometeor size distribution. To our knowledge this has not been done yet at least not for polarimetric variables at mm wavelengths.
In a cloud we have a variety of particles different in size, shape and orientation. Each individual particle produces a specific polarimetric response. For instance, some particles produce large

ZDR some produce small ZDR etc. These differences are defined by exact geometry of individual particles and this is not known. Therefore, one way to handle this issue is to characterize the particles independent of their size and fine shape since there is no way to get this from available observations. And this is what we do with the Rayleigh assumption. We introduce a parameter characterizing a general shape of particles but size and fine structure effects are ignored. When more information is available, for example from multifrequency observations, we can constrain more microphysical properties of particles, but with the current SLDR mode scanning cloud radar setup, only a proxy for a general shape can be constrained.

As reviewer raised a similar issue in question R1-C1, we kindly refer to our answer R1-A1.

R2-C7: L. 211: "footprint of spaceborne remote-sensing meteorological instruments" is a bit vague. Maybe refer directly to MW passive sensors (which also often measure polarimetric signals). Active MW sensors usually have better footprints and instruments in the VIS/IR as well.

R2-A7: Thanks for this remark. We rephrased Lines 225-226: "*e.g., to a footprint of a space-borne passive microwave sensor.*"

R2-C8 : L. 332-333: "the derived values of ξ should be analyzed with care when the method is applied to rain and the melting layer" I would add here also "close to the melting layer" as one can expect large aggregates forming there. And as you mention yourself in section 2.3 the Rayleigh assumption is not increasingly invalid for larger particles.

R2-A8: Thanks also for this remark. We rephrased lines 346-348: "*the derived values of ξ should be analysed with care when the method is applied to rain and close to the melting layer.*"

R2-C9: Section 4.1: You assume in this case that rain is always spherical. This is of course only true for small drops. I wonder whether you can see in the SLDR-RHI scan the effect of increasing oblateness of larger raindrops (especially when observed at low elevation angles)? I suggest to add maybe a comment to the text, for example, if this would be possible given the expected uncertainties of the observations.

R2-A9: Indeed, we occasionally observed effects of oblateness on the SLDR at low elevation angles at Limassol. We added a short text passage to lines 383-386 to highlight this possibility: "*With respect to the presented case it is noteworthy that it is likely that the observed rain droplets were small in size. This is corroborated by the absence of any elevation dependency of SLDR (Figure 9). In the case of strong rain, the oblateness of droplets would become apparent as SLDR increases from zenith pointing to 150° elevation angle, as we observed in some situations of moderate and heavy convective rain at Limassol during the CyCARE campaign.*"

Nevertheless, the effect of oblateness of large rain droplets basically is covered by the VDPS retrieval, since an increase of SLDR between 90° and 60° off-zenith is indicative of oblate particles, which is actually true for oblate rain droplets, as well.

Furthermore, it's important to note that we do not apply the VDPS method below the melting layer. It was carried out solely to provide an illustration pertaining to spherical particles such as droplets.

References:

Brdar, S. and Seifert, A.: McSnow: A Monte-Carlo Particle Model for Riming and Aggregation of Ice Particles in a Multidimensional Microphysical Phase Space, Journal of Advances in Modeling Earth Systems, 10, 187–206, https://doi.org/https://doi.org/10.1002/2017MS001167, 2018

Draine, B. T. and Flatau, P. J.: Discrete-Dipole Approximation For Scattering Calculations, J. Opt. Soc. Am. A, 11, 1491–1499, https://doi.org/10.1364/JOSAA.11.001491, 1994.

Kneifel, S., Kulie, M. S., and Bennartz, R. (2011), A triple-frequency approach to retrieve microphysical snowfall parameters, *J. Geophys. Res.*, 116, D11203, doi:10.1029/2010JD015430.

Kollias, P., Rémillard, J., Luke, E., and Szyrmer, W.: Cloud radar Doppler spectra in drizzling stratiform clouds: 1. Forward modeling and remote sensing applications, Journal of Geophysical Research: Atmospheres, 116, https://doi.org/https://doi.org/10.1029/2010JD015237, 2011.

Kneifel, S., Kollias, P., Battaglia, A., Leinonen, J., Maahn, M., Kalesse, H., and Tridon, F.: First observations of triple-frequency radar Doppler spectra in snowfall: Interpretation and applications, Geophysical Research Letters, 43, 2225–2233, https://doi.org/https://doi.org/10.1002/2015GL067618, 2016.

Leinonen, J., D. Moisseev, and T. Nousiainen (2013), Correction to "Linking snowflake microstructure to multi-frequency radar observations",J. Geophys. Res. Atmos.,118, 6708–6709, doi:10.1002/jgrd.50531.

Leinonen, J., Kneifel, S., and Hogan, R. J.: Evaluation of the Rayleigh–Gans approximation for microwave scattering by rimed snowflakes, Quarterly Journal of the Royal Meteorological Society, 144, 77–88, https://doi.org/https://doi.org/10.1002/qj.3093, 2018.

Lu, Y., Jiang, Z., Aydin, K., Verlinde, J., Clothiaux, E. E., and Botta, G.: A polarimetric scattering database for non-spherical ice particles at microwave wavelengths, Atmospheric Measurement Techniques, 9, 5119–5134, https://doi.org/10.5194/amt-9-5119-2016, 2016.

Matrosov, S. Y.: Polarimetric Radar Variables in Snowfall at Ka- and W-Band Frequency Bands: A Comparative Analysis, Journal of Atmospheric and Oceanic Technology, 38, 91 – 101, https://doi.org/https://doi.org/10.1175/JTECH-D-20-0138.1, 2021

von Terzi, L., Dias Neto, J., Ori, D., Myagkov, A., and Kneifel, S.: Ice microphysical processes in the dendritic growth layer: a statistical analysis combining multi-frequency and polarimetric Doppler cloud radar observations, Atmos. Chem. Phys., 22, 11795–11821, https://doi.org/10.5194/acp-22-11795-2022, 2022.

Welss, J.-N., Siewert, C., and Seifert, A.: Explicit habit-prediction in the Lagrangian super-particle ice microphysics model McSnow, https://doi.org/10.22541/essoar.168614461.18006193/v1, 2023

---

## Author Response (AR3)

**Response to the Editor**

Dear Editor,

We would like to express our gratitude to the reviewers for dedicating time to a third review and for providing valuable comments. The comments and suggestions from both Reviewers have been carefully addressed in the revised version of the manuscript. We once more acknowledge the concerns raised by both Reviewers regarding the Rayleigh scattering assumption which is underlying our retrieval. We aimed to provide concise replies to their statements, by basically summarizing what we argued in previous revision rounds and by implementing more text about our motivation and possible future enhancements into the manuscript.

Consequently, we would like to submit the revised manuscript and the diff-version of the revised manuscript together with our responses to all the comments provided by the two reviewers. In our replies, all references to modified lines are given with respect to the final markup/diff-version of the manuscript (with differences).

**Response to Reviewer #1:**

R1-C1: I am not satisfied with the response regarding the applicability of Rayleigh scattering at Ka band. I do see that you have discussed the characteristics of different scattering models in 2.3, but what I am questioning is the applicability of Rayleigh scattering at Ka band. The four points after "We decided to assume the Rayleigh scattering …" may be used to explain the spheroidal approximation when you are using S/C-band radars, and we know that the Rayleigh scattering is a good approximation at such bands. The critical point is the applicability of Rayleigh scattering at Ka band, but the authors did not discuss this. I hope the authors take this comment seriously, since the current draft left me the impression that we do not need to care about the non-Rayleigh scattering of ice at Ka band. If it is true, you should elaborate it. Otherwise, it is misleading to readers!

R1-A1: We apologize for giving Reviewer #1 the impression that we didn't consider his/her comment. It is right, that Rayleigh scattering is well applicable to C/S band (e.g, Dufournet et al., 2011; Melnikov and Straka, 2013) but recent studies suggest that there is only a modest influence of non-Rayleigh scattering on polarimetric variables (Matrosov, 2021). Matrosov (2021) notes that the possible reason for the smaller influence of non-Rayleigh scattering on polarimetric variables is that they are differential (rather than absolute) quantities representing differences/ratios of radar parameters at two orthogonal polarizations. Indeed, non-Rayleigh scattering seems to have a moderate impact using the VDPS method, as we discussed in R1-A2 of the previous revision cycle. We added the acknowledgement of cm-wavelength radars to Section 2.3 lines 159-172: "*It is well known that the Rayleigh approximation is not always applicable to simulate scattering from individual and large ice particles at wavelengths shorter than C-band, which holds especially for absolute values such as reflectivity factor (Lu et al., 2016). At shorter wavelengths, the direct dipole approximation (DDA, Draine et al., 1994) can be used to simulate scattering of individual ice particles having a complex shape. Meanwhile, extensive databases exist (Lu et al., 2016) and found, e.g., special attention already for the application of multi-wavelength radar studies (Von Terzi et al., 2022). However, these simulations and associated studies are often limited to a number of predefined shapes and therefore do not necessarily represent the realistic distribution of ice particles observed by a radar (Leinonen et al., 2018). Simulations for a single particle also do not reflect the volumetric scattering effects of a large population of hydrometeors. In general, ice particles in a scattering volume have arbitrary shapes and the contribution of individual particles to the backscattering radar observables and especially polarimetric quantities is averaged out (Matrosov, 2021; Von Terzi et al., 2022). We decided to assume the Rayleigh scattering and the spheroidal particle approximation (Matrosov et al., 1991, Ryzhkov et al., 2001, Bringi et al., 2001) because (1) such a model explains general polarimetric scattering effects*

*with just a few parameters (axis ratio, permittivity and canting angle), (2) the model parameters are well constrained by the observations, (3) the volumetric scattering is taken into account, and (4) the model allows a computationally effective derivation of the polarizability ratio."*

R2-C2: In the conclusion section, I do not appreciate the added discussion on the "shortcomings of the Rayleigh approximation". You did not discuss the shortcoming of using Rayleigh approximation, please elaborate the SHORTCOMINGS!

R1-A2:  As requested, we extended the discussion of shortcomings in Section 2.3 (lines 159-172), See R1-A1), and in the Conclusions Section. We propose to add in the Conclusions Section Lines 513-523: "*Finally, in our study, we are assuming Rayleigh scattering and describe the particle shapes according to the aspect ratio and the permittivity. In reality, ice crystal shapes are more complex and need a more sophisticated scattering method to accurately capture scattering of particles with axis lengths exceeding the range of the Rayleigh scattering regime. This holds definitely true for absolute quantities such as reflectivity at wavelengths shorter than C-band (Lu et al., 2016, Von Terzi et al., 2022). However, a recent study of Matrosov (2021), demonstrates that the influence of non-Rayleigh scattering is weak for polarimetric variables such as LDR. As a likely reason for this behaviour, Matrosov (2021) hypothesizes that polarimetric variables are differential (rather than absolute) quantities representing differences/ratios of radar parameters at two orthogonal polarizations.*"

R1-C3: Regarding the definition of isometric particles, the authors say that "non-spherical particles with low density (low refractive index) also appear to be isometric". The logic is awkward. Aggregates with low AR are expected to have polarizability ratio of ~1 due to low density, but they are not isometric.

R1-A3:  We did not intend to say that particles with low refractive index are isometric. What we mean is that certain particles may exhibit characteristics that make them appear isometric when observed with a radar, although they are not by geometric definition. Reason is that besides the geometric axis ratio, also the permittivity plays a role in the determination of the polarizability ratio. In the case of particles with a low refractive index (i.e., low permittivity), their reduced response to radar waves may lead to scattering characteristics that resemble those of isometric particles. We added Lines 178-180: "In *the case of particles with a low refractive index (i.e., low permittivity), their reduced response to radar waves may lead to scattering characteristics that resemble those of isometric particles*"

**Response to Reviewer #2:**

I like to thank the authors again for addressing my comments and concerns. I support the publication of the current article and don't want to delay it for another round of reviews and discussion. However, I am honestly a bit disappointed that I could not find answers to two of my main open questions regarding this undoubtedly promising and useful technique:

R2-C1: I cannot really agree with the author's in their argumentation related to the scattering properties: Of course, the radar always measures an average of the single scattering properties of all particles present in a certain radar volume. But how can you be sure the spheroidal Rayleigh approximation is indeed a good approximation for it? I also don't see why a comparison of the spheroidal method with simple solid-ice hexagonal ice plates or columns would be such a problem. Yes, the experiment would not be sufficient to answer whether the spheroidal approximation is accurate for the volume scattering properties, but it would shed light on the question how good the method is for specific situations (distinct shape). If this experiment reveals that the two methods agree well for specific shapes and configurations (orientation etc.), the assumption that the spheroidal method is indeed a good approximation for the volume scattering properties would be strongly affirmed.

R2-A1: Given the concerns raised by both reviewers, we acknowledge that the response of polarimetric parameters to non-Rayleigh scattering is a pressing topic. Unfortunately, this topic is out of the scope of the current study. The intention of our study was to bring the original STSR approach of Myagkov et al. (2016) to the broader community of SLDR radar application. For our study we base our assumption on the applicability of the VDPS method to non-Rayleigh targets on findings from other recent studies, such as the one of Matrosov (2021). As we discussed in R1-A2 of the previous revision cycle, non-Rayleigh scattering seems indeed to have only a moderate impact using the VDPS method. Please also note that we provide further responses to a similar question which was already raised by Reviewer 1 in R1-C1 and R1-C2. We thus kindly refer to our response R1-A1 and R1-A2. We added Lines 524-525 : "*If these are applied to realistic hydrometeor populations, a model-based validation of the hypothesis of Matrosov (2021) shall be feasible.*".

R2-C2: As the question regarding scattering assumptions are still open, a comparison with in-situ observations would be in my view even more important. Certainly, also in-situ methods have their uncertainties, but it would provide the possibility for a direct comparison of retrieved and real particle shape properties. At the moment we can neither be sure about the accuracy of the scattering properties nor about the uncertainty due to other assumptions, such as "same particles are present in the main spectra peak".

R2-A2: Yes, thank you for this remark. It is planned to compare the VDPS method with in situ observations in a currently ongoing campaign in Switzerland to validate the method. Results from this campaign will be available only by 03/2024.

Technical corrections :

R2-C3: L. 178: Add Figure before "2a"

R2-A3: We added Line 180: "Figures"

**References:**

Dufournet, Y. and Russchenberg, H. W. J.: Towards the improvement of cloud microphysical retrievals using simultaneous Doppler and polarimetric radar measurements, Atmos. Meas. Tech., 4, 2163–2178, https://doi.org/10.5194/amt-4-2163-2011, 2011.

Matrosov, S. Y.: Polarimetric Radar Variables in Snowfall at Ka- and W-Band Frequency Bands: A Comparative Analysis, Journal of Atmospheric and Oceanic Technology, 38, 91 – 101, https://doi.org/https://doi.org/10.1175/JTECH-D-20-0138.1, 2021.

Melnikov, V. and Straka, J. M.: Axis ratios and flutter angles of cloud ice particles: retrievals from radar data, J. Atmos. Ocean. Tech., 30, 1691–1703, doi:10.1175/JTECH-D-12-00212.1, 2013.